# Circular bioeconomy in African food systems: What is the status quo? Insights from Rwanda, DRC, and Ethiopia

**Haruna Sekabira**[1]\*, **Elke Nijman**[1], **Leonhard Späth**[2,3], **Pius Krütli**[3], **Marc Schut**[1,4], **Bernard Vanlauwe**[5], **Benjamin Wilde**[2], **Kokou Kintche**[6], **Speciose Kantengwa**[1], **Abayneh Feyso**[7], **Byamungu Kigangu**[6], **Johan Six**[2]

1 Division of Natural Resources Management, International Institute of Tropical Agriculture, Kigali, Rwanda, 2 Department of Environmental Systems Science, Sustainable Agroecosystems, ETH Zurich, Zurich, Switzerland, 3 Department of Environmental Systems Science, Transdisciplinarity Lab (TdLab), ETH Zurich, Zurich, Switzerland, 4 Knowledge, Technology and Innovation Group of Wageningen University, Wageningen, The Netherlands, 5 Division of Natural Resources Management, International Institute of Tropical Agriculture, Nairobi, Kenya, 6 Division of Natural Resources Management, International Institute of Tropical Agriculture, Kalambo, DR Congo, 7 Department of Agribusiness and Value Chain Management, Arba Minch University, Arba Minch, Ethiopia

\* H.Sekabira@cgiar.org

**Data Availability Statement:** All relevant data are within the paper and its Supporting Information files.

## Abstract

Increasing global food insecurity amidst a growing population and diminishing production resources renders the currently dominant linear production model insufficient to combat such challenges. Hence, a circular bioeconomy (CBE) model that ensures more conservative use of resources has become essential. Specifically, a CBE model that focuses on recycling and reusing organic waste is essential to close nutrient loops and establish more resilient rural-urban nexus food systems. However, the CBE status quo in many African food systems is not established. Moreover, scientific evidence on CBE in Africa is almost inexistent, thus limiting policy guidance to achieving circular food systems. Using a sample of about 2,100 farmers and consumers from key food value chains (cassava in Rwanda, coffee in DRC, and bananas in Ethiopia), we explored existing CBE practices; awareness, knowledge, and support for CBE practices; consumers' opinions on eating foods grown on processed organic waste (CBE fertilizers), and determinants of such opinions. We analysed data in Stata, first descriptively, and then econometrically using the ordered logistic regression, whose proportional odds assumption was violated, thus resorting to the generalized ordered logistic regression. Results show that communities practice aspects of CBE, mainly composting, and are broadly aware, knowledgeable, supportive of CBE practices, and would broadly accept eating foods grown CBE fertilizers. Households with heads that used mobile phones, or whose heads were older, or married, or had a better education and agricultural incomes were more likely to strongly agree that they were knowledgeable and supportive of CBE practices and would eat CBE foods (foods grown on processed organic waste). However, the reverse was true for households that were severely food insecure or lived farther from towns. Rwandan and Ethiopian households compared to DRC were less likely to eat CB foods. Policies to stimulate CBE investments in all three countries were

**Funding:** This work has been funded by the Swiss Agency for Development and Cooperation (SDC) for funding RUNRES under grant no: 7F09521. The funders had no role in study design, data collection and analysis, decision to publish, or preparation of the manuscript.

**Competing interests:** The authors have declared that no competing interests exist.

largely absent, and quality scientific evidence to guide their development and implementation is currently insufficient.

## 1. Introduction

The world is increasingly challenged to sustainably feed its ever-increasing population amidst diminishing resources and worsening climate change impacts [1]. Therefore, without sustainable production and consumption systems, the achievement of several of the strategic sustainable development goals (SDGs), like poverty alleviation, food security, environmental health, and sustainable cities is in jeopardy [2–4]. Moreover, part of the causes for the worsening global food production and consumption systems' situation is humans, whose dignified existence is the prioritized aim of the SDGs. For instance, the linear model of resource use employed by modern food production and consumption systems is blamed for depletion of resources [3,5]. Under this linear model, production of food occurs in rural areas, while most consumption occurs in urban areas, where most recyclable organic waste (food, household, and human) accumulates in uncontrolled dumpsites, rudimentary sanitation facilities, or is released into the environment, particularly in the least developed countries [4,6–12]. However, organic waste (*any biological waste from farm and other green residues, food, household, processing plants, or excreta from both livestock and humans that can easily be recycled naturally by microorganisms*) contains valuable soil nutrients, and not reusing them in food systems where they were mined breaks nutrient loops, which results in long term soil nutrient depletion [13]. Indeed, the globe is ultimately circular in use of her resources, however, if resources are mined heavily and the waste or biproducts from these resources are not reused in the particular food system where production was done, then the resource use model is a linear one [3,6]. Such a linear production system renders the rural-urban food system nexus non-resilient. Moreover, accumulation of waste in urban areas constitutes an environmental concern that threatens both human and environmental health [14]. In essence, the linear model of modern food production is vulnerable to climate change effects, and contributes to food insecurity, poverty, poor human health, soil degradation, and biodiversity loss–hence it is unsustainable [3,4,10].

On that basis, global efforts have recently focused on shifting from a linear model of resource use to a circular bioeconomy (CBE) one. A CBE model reduces waste produced across a given supply chain, utilizes food waste, and all related organic waste within the chain (human excreta, farm residues, waste water, food processing waste, and others) to recycle nutrients [3,4,15,16]. Circular bioeconomy is a combination of circular economy and bioeconomy [3]. A circular economy aims to rearrange the linear take–make–use–dispose resource use model in production and consumption systems with a circular formation [6], while bioeconomy aims to provide goods and services sustainably through use of biological resources, processes, and products [3,4]. Under the CBE food system, production would still mostly occur in rural areas, and consumption in town centres. However, after urban consumption, organic waste (food, household, and human) would be collected, recycled, and returned to rural areas for reuse on farms [12,17]. Because caution is undertaken, under the CBE model to protect and restore the healthiness of production ecosystems; the CBE model in its entirety differs from the usual mere recycling of waste [4]. In fact, Carus and Dammer [15], assert that a CBE model aims to provide renewable biological resources and converts such resources and their waste streams into products of added value(food, animal feed, bioenergy, or others), while creating new employment opportunities. Therefore, reusing biologically recycled

biomass and other waste in areas of a particular food system, where production occurs, is what makes such a food system a CBE one [4]. Reusing recycled waste in this manner would close nutrient loops, replenish soil nutrients and soil organic matter, all of which would increase sustainable farm productivity [13]. More so in African food systems that are characterized by low farm inputs, there is potential to transform organic waste into useful farm inputs [18]. Increased crop or animal productivity from use of recycled nutrients (a*ssuming low production costs of efficiently processed inputs (recycling done in homes or at dumpsites to avoid transporting bulky municipal organic waste expensively from dumpsites to homes, or original waste is sorted at source to minimize additional costs in sorting), since organic waste is abundantly available*) could also increase household incomes via improved crop and animal yields or reduce household expenditure via low food costs caused by increased food supply. Thus, poverty as well as other household welfare challenges (education, health, nutrition, housing, communication etc.) could be addressed via this income pathway. The CBE model would also ensure that waste does not become an environmental problem in towns, where it harbours pathogens that threaten human health, emits greenhouse gases (GHG), or pollutes air, and thus lower the environmental quality necessary for proper human health [12,19]. The use of organic compost from recycled waste could also reduce the need for inorganic fertilizers, thus reducing costs on buying fertilizers [20].

However, most farmers and consumers in Africa are familiar with the linear model (produce food, and take it to consumption areas like towns and leave the waste to be disposed of there without being reused where food was produced), and view organic waste as valueless [19,21]. Furthermore, in some communities, even in rural areas, it is perceived as against social and cultural norms to harvest waste as a resource, especially human waste [22]. Nevertheless, a CBE model must have an approach backed by local policies, that is compatible with key social, cultural, ethical, and ecological frameworks in order to achieve a sustainable pathway [3,23]. Sadly, policies in most African countries do not prioritize organic waste recycling and reuse, and in some countries, CBE policies are largely inexistent [10,24]. Further problematic is that scientific evidence is currently insufficient to guide global food systems' policies on addressing gaps around CBE [6,16,24]. Despite good CBE prospects, actors in food systems need to understand the potential of CBE concepts, and how these changes can be effectively implemented [1]. With our study, we contribute to enabling food systems' actors in studied African city-regions to understand the local communities' status quo on CBE practices (*recycling of organic waste (food, farm, green, household, excreta (livestock and human)) for reuse as compost, biochar, livestock feed, cooking energy pellets, and others*) vis-à-vis community attitudes, perceptions, and preferences. Moreover, Geissdoerfer et al. [6] note that, to date, the principle focus of CBE research has been on developed countries dominated by (in decreasing importance, China being the most researched): China, UK, Netherlands, USA, Italy, Japan, Sweden, Germany, Belgium (up to 25 western countries). Africa was not featured in recent developments of CBE, perhaps because severe consequences of the linear model have been most pronounced in highly industrialized countries.

However, the rapidly increasing African urban population is as well exposed to the severe consequences of food insecurity and poor sanitation in urban centres due to waste accumulation and excessive soil nutrients depletion in rural areas [10,11]. If this non-restorative use of resources is not reversed, sustainable development in Africa grounded on efficient resource-use food production and consumption systems could be difficult to attain [4,9,10,24,25]. Therefore, we contribute towards sustainable food systems by analysing CBE practices in three African city regions based on data from 2,175 households. This data has been collected by the project "Rural-Urban Nexus: Establishing a Nutrient Loop to Improve City Region Food Systems Resilience (RUNRES; https://runres.ethz.ch/)", funded by the Swiss Development Agency

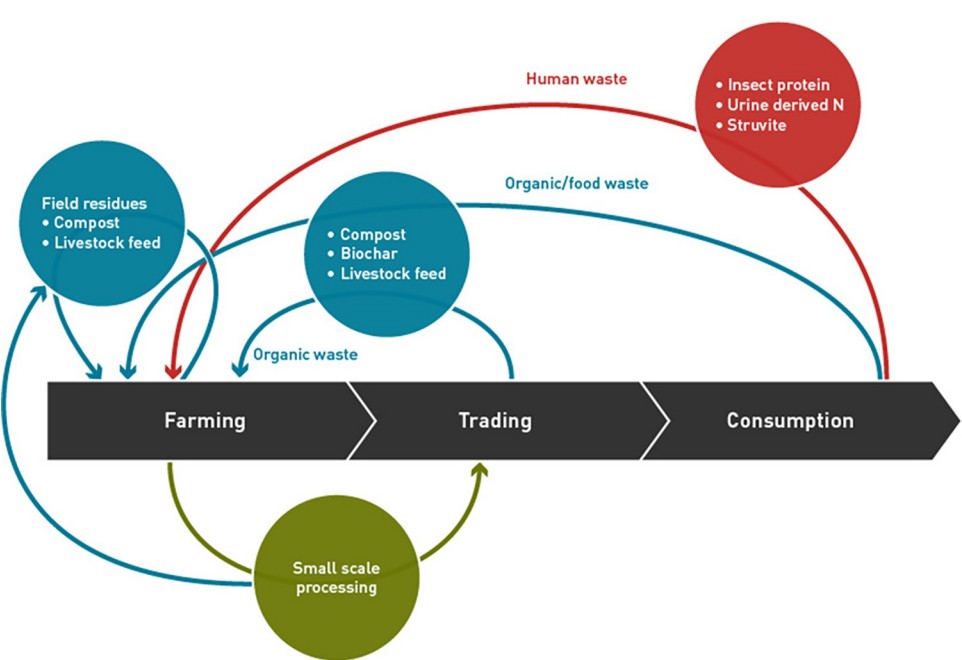

**Fig 1. The envisioned circular bioeconomy (CBE) to be established through CBE innovations co-designed and co-implemented by RUNRES.**

(SDC). RUNRES aims to establish a circular bioeconomy in 4 African city regions by using cost-effective, and socially acceptable CBE innovations to close nutrient loops through valorisation of organic waste in food systems. RUNRES aims to map strategic food commodity value chains, and all their associated organic waste streams, and then co-design with local stakeholders, CBE innovations that can valorise this organic waste (household, farm residues, food, and human) into products (compost, biochar, and livestock feed) reusable in agriculture while as well it enhances social welfare through creation of employment opportunities for especially the youth and the women who are the dominant actors in waste streams. Therefore, RUNRES works with municipalities, waste collection companies, farmer organizations, food processors, local and national governments to realize this goal. In Fig 1, we show as adopted from RUNRES [11], a graphical presentation of the envisioned CBE by RUNRES.

More specifically, RUNRES currently operates in four African city regions, focusing on four food commodity value chains: cassava in Kamonyi (Rwanda), coffee in Bukavu (Democratic Republic of Congo (DRC)), bananas in Arba Minch (Ethiopia), and vegetables in Msunduzi (South Africa). As, our contribution to the general CBE literature, we focus only on a subcomponent of RUNRES, the social perceptions of communities especially consumers regarding their awareness, knowledge, and support of CBE practices generally, and acceptance of CBE products, specifically foods grown on different types of organic residues (treated urine, faecal matter, and other livestock or green waste (compost)). We used data from all city regions, except Msunduzi, South Africa (due to data unavailability) to investigate CBE status quo in these city-regions. Specifically, we sought answers to the following questions:

1. What CBE practices do currently exist in the respective city-regions' food systems?

2. Are communities currently aware, knowledgeable, or supportive of CBE practices?

    a. What factors do explain such CBE awareness, knowledge, and support?

3. What are communities' opinions on consuming foods grown on CBE fertilizers?

   a. What factors do explain these consumption opinions?

4. Are there policies guiding CBE practices in the selected city-regions?

   a. If yes; what are these policies' guidance on CBE practices?

Addressing research questions above, generated knowledge on where we are with CBE practices in the mentioned city-regions. This knowledge would aggregately guide efforts on successfully designing and scaling of economically suitable, socially, and legally acceptable CBE innovations. Economic suitability, social and legal acceptability are attributes that render CBE innovations desirable and sustainable [3,4,10,11,15]. For information on CBE fertilizers, first, we consider compost in a way most known to communities, that is, compost from household, farm or green residues or animal manure, but because urban centres are chocking on increasing human populations, hence we also look at human excreta differently as those fertilizers made from treated urine, or faecal matter. The differentiation of human excreta from the general compost was based on the anticipated negative opinion of communities on reusing human waste, and their lay understanding of compost as only that made from green, farm, and livestock residues.

The rest of the paper is presented as follows: next, we present the methods and data used for this study, after which we present results and then discuss them separately. Finally, we conclude with concise guidance on policy areas.

## 2. Methods and data

### 2.1 The circular bioeconomy concept

For this study, we adopted the circular economy (CE) definition of Kirchherr et al. [25], which broadly defines the CE concept as "a system that replaces the 'end-of-life' notion, through a combination of principles and practices that not only reduce waste, but recycle and reuse available waste. However, literature also takes note of multiple and somewhat divergent interpretations of the CE concept [8,25]. In fact, Kirchherr et al. [25] adds that some authors confuse the CE concept with just recycling, while he argues that the CE concept must include design elements that reduce waste, recycle it, and reuse it efficiently. However, CE has been linked with bioeconomy which broadly refers to the use of biological resources, processes, and products to sustainably produce consumables [3]. Therefore, linking the two, a CE focusing on the reduction and reuse of organic waste from food supply chains has been called a circular bioeconomy (CBE) [3,15]. More specifically, El-Chichakli et al. [26] expounded on this by making it clearer that a circular bioeconomy is a model that recycles and reuses organic waste through biological processes like fermentation, biotechnology, and molecular biology. Not reusing organic and waste products in a food system, where production takes place renders a system linear and breaks the nutrients cycle because decomposing waste for instance in towns doesn't avail these nutrients to be reused in food production in rural areas [4]. Therefore a CBE model must ensure an efficient cycle that biologically recycles and reuses products and waste resources after consumption back in food production to render the system circular [15]. Given the focus of RUNRES on the capture, biological processing, and reuse of available organic waste in waste streams in the four-city regions, we focus on CBE in this study.

### 2.2 Research area, commodities, sampling, surveying, and data

Our work was done in three different African countries where RUNRES is promoting CBE based innovations. However, notice must be made that data analyzed here were collected

before RUNRES started implementing CBE innovations to properly understand CBE status quo. The exact locations within these countries were called RUNRES city-regions which were purposively selected because of their dominance in growing the food commodities of interest, and their existing connections with project partners and stakeholders. These city-regions and their respective food commodities were Kamonyi of Rwanda for cassava, Bukavu of DRC for coffee, and Arba Minch of Ethiopia for bananas. Once city-regions had been selected, we then identified the most relevant food commodities within each city-region, through preliminary and participatory work, with our stakeholder networks [11]. Our selection was participatory in that all RUNRES stakeholders from the private, academia, public sector, funders, and local communities willing to participate, took part in designing the RUNRES approach, food commodity value chains, and fitting innovations via transdisciplinary workshops organized by the RUNRES team. The main project private sector partners in all countries were garbage collection companies, while main public partners were municipal authorities. Within each city region, we selected various administrative sub-divisions randomly (we listed all sub-divisions from each city-region, and randomly selected those we worked with). We then selected respondent households through a systematic random sampling of available household lists from those respective sub-divisions. In Rwanda, we obtained household lists from sector (sub-division) offices for farmers and consumers. For farmers, and consumers, whose numbers would be generated sufficiently, we chose a household at every fifth interval in Rwanda, sixth interval in Ethiopia, and seventh interval in DRC to be surveyed. These intervals were used to avoid a sample dominated by certain villages/cells (smallest administrative unit) within the city-region–thus the selections were differently probabilistic. In Ethiopia, household lists were obtained from *Kabele* (sub-division) offices for both farmers and consumers. While in DRC, a farmers' list was obtained from the national coffee office in Bukavu and validated by coffee cooperatives. A consumers' list was obtained from Bukavu municipality offices and validated by sanitation service providers. Other value chain actors like inputs sellers, middlemen, processors, wholesalers, and retailers, were identified in a snowball sampling methodology beginning with farmers. This was because these actors were very few and were sparsely distributed. Our study was granted the ethics approval by the Rwanda National Ethics Committee (RNEC)–composed of 9 of its11 members. The committee stated that, *"After a review of the protocols and consent form (for respondents' choice to participate or not), during the 10th of October 2020 RNEC meeting where quorum was met, and revisions made (on earlier protocols and consent forms) on the advice of RNEC submitted on 28th June 2021, we hereby provide approval for the protocols, valid for 12 months"*. RUNRES's work and protocols is also covered by the ethics clearance of the Ethics Commission of ETH, Zurich, under reference No. EK 2020-N-51 (approval attained June 2020). ETH Zurich is the overall implementing institution for the RUNRES project.

The survey protocol was tested on a trial sample in the target city-regions to validate that respondent could understand the questions as intended, and after identification of the respondents, we then administered the survey to final respondents. Each respondent category (farmers, consumers, processors, etc.,) would have an independent section within a common survey tool with closed and open-ended questions (see S1 File). Our interviewees were the household heads responsible for principally for household decisions, and where they were unavailable the spouse, who is usually the secondary decision maker, was interviewed. Following WFP [27] protocols for representative samples in food commodity value chain analysis, we set 400 households per value chain actor segment as our target sample size. However, from field explorations, we found that some actors' segments, were either non-existent or had far much less than 400 households. Numbers dwindled more when we executed systematic random sampling for participants. However, because of a high population density in Rwanda we were able

to have a random sample of over 400 households for farmers and consumers. We used Rasoft software from http://www.raosoft.com/samplesize.html to confirm WFP sample recommendations if we aimed at a 95% confidence interval and a 5% error margin vis-à-vis city-region farmer populations. Subsequently, we interviewed over 2,100 households (farmers and consumers) across all three countries, using about 2 months in each of Rwanda and DRC, and about 6 weeks in Ethiopia. We analysed data using Stata/SE 16.0. However, other value chain analyses including key activities and how they are executed, challenges and opportunities at each actor segment, and market margins are presented in a technical report that we presented to the project steering committee and available upon request.

We collected data electronically using open data kit (ODK) computer packages between August and December 2019 with the help of trained enumerators who were graduates from local universities in fields related to agricultural sciences. Enumerators could speak at least both English and the local language, and this enabled enumeration in local languages while data inputs were in English. Questions ranged from household biodata, demographics, agricultural production, food consumption, waste generation, income, to social attitudes and perceptions on CBE practices. Some questions were specific to certain actor segments. However, most questions, such as those on household demographics, consumption, awareness, knowledge, support for CBE practices, and opinion on eating CBE foods, were asked to all respondents.

Having an interest in understanding the CBE status quo with regards to policies, we also collected some policy data by reviewing potentially CBE related policies from government websites using keywords such as agriculture, environment, organic farming, composting, waste management etc. We cross-checked specific country policies using expert and key informant interviews administered by locally resident RUNRES scientists to our local partners, municipal authorities, and ministry staff including that of environment, agriculture, standards, industry, and trade. From local reviews added to online reviews, we realized that CBE policies are largely unavailable (DRC) and where available, were largely incomprehensive (Rwanda and Ethiopia). Hence, we did not investigate policies further but briefly comment on them here. For instance, in Rwanda, there is policy advocacy for organic composting, but use of human waste is prohibited due to potential health risks [28,29]. In Ethiopia, policies promote the use of organic matter to improve soil structure and fertility. Utilization of on-farm forage, and fodder as animal feeds is also promoted. However, there are no regulations on use of human waste as fertilizer [30,31]. We present a summary of those CBE related policies we became aware of from both online searches, experts, and key informant interviews in supporting information, S1 Table. We also present the minimally anonymous data used for this paper in supporting information data file, S2 File. Because some actor segments, had fewer numbers of respondents, as a natural limitation, their samples could not be used to draw meaningful statistical inferences across all countries; hence, we do not use these other actors in our empirical analysis. Nevertheless, we use some of the observations and lessons learned from these actors, to present and discuss empirical results in this study, and draw plausible recommendations. We present all interviewed value chain actors in Table 1.

## 2.3 Data analysis

**2.3.1 Descriptive analysis.**   We have used means or percentages of various variables generated in Stata to descriptively describe the status quo around CBE practices and CBE foods in target countries. The descriptive analysis gives us a summarized general data outlook on key outcome and explanatory variables.

**2.3.2 Regression analysis.**   Because for instance, we are interested in understanding households' varying preferences (that are intrinsic but presented in a numerically meaningful

**Table 1. Sample sizes across countries.**

| Actor type | Rwanda (1,160) | DRC (809) | Ethiopia (593) |
|---|---|---|---|
| Inputs stockists | 25 | 7 | 0 |
| Farmers | 520 | 281 | 190 |
| Middlemen / traders | 61 | 94 | 10 |
| Processors | 23 | 50 | 1 |
| Wholesalers | 12 | 22 | 0 |
| Retailers | 54 | 38 | 0 |
| Consumers | 465 | 314 | 392 |

way) for various CBE foods, based on assumptions that consumers make about these CBE foods, like safety, cultural acceptability, and others, we follow Fishburn [32] and rely on the utility theory to model these preferences given a set of choices. More specifically, since our set of choices were merely rankings, that could only tell us what a consumer preferred more or less but not necessarily telling us the value attached to such preference, we assume the ordinal utility theory [33,34]. The ordinal utility theory assumes that it is worth to only ask which choice is better than the other, but it is worthless to ask how much better a choice is or how good it is compared to others [34]. Therefore, we follow an ordinal utility function to model farmers and consumers' preferences that are presented on an ordinal scale (strongly agree, agree, neutral, disagree, and strongly disagree). We do not derive the ordinal utility function to minimize space, but simply apply the concept. Hence, we subsequently estimated Eq (1), as an ordered logistic regression since outcomes were ordinal, to enable us understand the likelihood of each ordinal outcome category, and the factors that could explain such outcomes [35,36]. Generally, we used the logistics regression because our data was reflecting discrete choices of households using these CBE practices, and logistic regressions are comparatively easier to interpret [36].

$$AKSF_i = \alpha + \beta X_i + \partial Z_i + \varepsilon_i \qquad (1)$$

$AKSF_i$ was the ordinal outcome variable depicting at what level (strongly agree, agree, neutral, disagree, and strongly disagree) were households aware, knowledgeable, or supportive of CBE practices, or at what level would households accept to eat foods grown on compost, treated urine, or faecal matter. $X_i$ is a vector of household characteristics, and $Z_i$ is a vector of contextual factors including country effects; while $\alpha$, $\beta$ and $\partial$ are parameters we estimated. $\varepsilon_i$ is a random error. However, proportional odds logistic regression models as the ordered logit, must satisfy a key fundamental assumption requiring that no explanatory variable has a disproportionate effect on a particular outcome variable level / category. This is the proportional odds assumption. On testing data for this assumption as a validation step for our model, we found that our data violated this assumption, thus we re-estimated Eq (1) as a generalized ordered logistic regression to correct this violation [37]. Test results are presented below each results Table.

In our analysis, we selected as explanatory variables age, education, agricultural income, distance to nearest major town, severe food insecurity, marital status, gender, household size, and mobile phone use. Our choice is not only informed by field observations but also by literature, which we do not cite here to avoid redundancy, since we dominantly use the same in the discussions. Such literature and field observations guided us to the following general hypotheses: 1) households that are food insecure, married, or educated are likely to support CB practices, and willing to eat CB foods, 2) urban households are not likely to support CB practices, nor willing to eat CB foods, because there is currently heavy accumulation of waste in towns

**Table 2. Description of variables used in regressions.**

| Countries | Means | | | | | |
|---|---|---|---|---|---|---|
| | DRC (N = 595) | | Rwanda (N = 997) | | Ethiopia (N = 583) | |
| Distance to nearest major town (km) | 15.2 | (15.7) | 5.75 | (5.61) | 11.25 | (7.77) |
| Education of head (years) | 8.04 | (5.99) | 6.29 | (3.91) | 5.81 | (5.52) |
| Household size (Persons) | 7.69 | (3.31) | 5.12 | (1.98) | 5.75 | (2.37) |
| Agriculture income/season (USD) | 134.9 | (217.2) | 151.4 | (275.4) | 466.8 | (936.1) |
| Age of head (years) | 44.4 | (15.7) | 45.4 | (12.1) | 44.91 | (16.6) |
| Severe food insecurity (score from 3 questions) | 0.89 | (1.07) | 0.35 | (0.84) | 0.09 | (0.42) |
| *Percentages (%)* | | | | | | |
| Male household head (D) | 74 | | 77 | | 79 | |
| Mobile phone use (D) | 57 | | 92 | | 78 | |
| Severe food insecurity experienced (D) | 51 | | 18 | | 5 | |
| Household head is married (D) | 82 | | 85 | | 83 | |

In parentheses are standard deviations; USD is United States Dollar, 1USD = 1,607 Congolese francs = 930 Rwandese francs = 30 Ethiopian birr; D is dummy; Severe food insecurity refers to questions 7, 8, and 9 of (Household Food Insecurity Access (HFIAS) protocol.

[38,39]. In addition, because waste management is labour intensive [38,39], we hypothesized that married or larger sized households would support CBE practices, since they would have higher person numbers for labour. We present these variables in Table 2.

For the regression results' interpretations, we prioritized associations of covariates for the strongly agree category, because interpreting all categories' results would be so bulky. Moreover, the strongly agree category is a very high standard that could more likely yield predictions of a higher possibility to be valid internally and externally. Nevertheless, we present all other results in respective Tables for readers' comparison. For continuous variables that showed a non-normal distribution, we followed Box and Cox [40] to transform the affected variables, to yield residuals that were less heteroskedastic and approximately distributed normally. Boxcox transformations also corrected for average outliers, but for extreme outliers—for instance a respondent who appeared to have over 120 years, these were followed up physically through telephoning respondents for the correct data. Outliers could easily be detected with the tab command in Stata.

Regarding outcome variables, we use the terms based on a generalized perception of respondents–where awareness was merely having heard of such a CBE practice, and knowledge was having any skill about how that CBE practice could be realized, and support was generally reflecting an opinion for if respondent would accept / recommend such a CBE practice to other community members. Because we focused on households that were cautiously using biological products and processes like fermentation to recycle and reuse organic waste (food, household, and human) with an intent of reducing waste and closing nutrient loops within their own food systems and improve their food systems' sustainability and productivity, we classified these practices as CBE ones, and their food products as CBE foods, following literature [3,4,6,15].

## 3. Results

Briefly, we generally describe the sample basing on explanatory variables used and present these results in Table 2. From Table 2, across all three city-regions, we observed that households are mostly male headed, married, with heads mostly attaining a primary school level for formal education. Household heads were, on average aged 45 years across the three city-

**Table 3. Current circular bioeconomy (CBE) practices.**

| Country | Actor type | CBE practice (production and use of) | Percentage | Source of materials | Destination of CBE product |
|---------|-----------|--------------------------------------|-----------|---------------------|---------------------------|
| Rwanda | Farmers | Organic compost, animal feeds | 77 | Crop residues, cassava peels, and animal excreta | Crop production, livestock feeds |
| DRC | Farmers | Organic compost | 99 | Crop residues like coffee husks and animal excreta | Crop production |
| | Consumers | Organic compost, and animal feeds | 3 | Household organic, food, and human (urine) waste | Crop production and poultry feeds |
| Ethiopia | Farmers | Organic compost | 75 | Crop residues and animal excreta | Crop production |

regions. Household sizes ranged from 5 persons in Kamonyi-Rwanda to 8 in Bukavu-DRC. Most households used mobile phones. Agricultural income in Arba Minch-Ethiopia was nearly three times that of samples in Kamonyi or Bukavu. About half of the households in Bukavu experienced a form of severe food insecurity, while only 18% in Kamonyi and 5%in Arba Minch shared a similar experience. High severe food insecurity in Bukavu may be due to persistent armed conflicts in eastern DRC, where our sample area belongs [41].

## 3.1. Descriptive results as per research questions

**3.1.1. What CBE practices do currently exist in the respective farming systems?.** In all three countries, some CBE practices are prevalent (Table 3). For instance, most Rwandan farmers (77%) produce organic compost from crop residues and animal excreta and used this in crop production. In DRC, almost all farmers (99%) produce and use organic compost, but only about 3% of consumers recycle waste and reuse it as compost or animal feed in urban agriculture. Consumer samples in all other countries never practiced CBE activities. In addition, most (75%) farmers in Ethiopia recycle crop residues or animal excreta and reuse it as organic compost.

**3.1.2. Are communities currently aware, knowledgeable, supportive of CBE concepts?.** Most respondents stated that they were aware of, knowledgeable of, and/or supportive of CBE practices (Table 4). The least (55%) sample proportion for agreement (agree and strongly agree categories) to any CBE aspect (awareness, knowledge, or support) was in DRC, whereas this proportion was as high as 89% in Rwanda.

**3.1.3. What are communities' opinions about consuming foods grown on CBE fertilizers?.** The stated acceptance to eat CBE foods is quite high across countries and more so for foods grown on organic compost (Fig 2). In Rwanda, about 91% would accept (agree + strongly agree) to eat CBE foods from organic compost, treated urine, or faecal material. These proportions are also relatively high in DRC (84%), and Ethiopia (72%). However, on eating CBE foods from human waste, there was disagreement (disagree + strongly disagree) of at least 8% across all countries. Moreover, sample proportions that were neutral (indifferent to accept or not accept) were also high, reaching about 22% in Ethiopia for CBE foods from urine and faecal fertilizers. In Fig 2, we show these opinions on a 5-level scale (strongly agree, agree, neutral,

**Table 4. Household awareness, knowledge, or support for CBE practices.**

| | Percentages (%) | | | | | | | | |
|---|---|---|---|---|---|---|---|---|---|
| **Countries** | **Rwanda (N = 997)** | | | **DRC (N = 595)** | | | **Ethiopia (N = 583)** | | |
| Variables | Aware | Knowledge | Support | Aware | Knowledge | Support | Aware | Knowledge | Support |
| Strongly agree | 16 | 14 | 21 | 17 | 17 | 16 | 39 | 32 | 31 |
| Agree | 72 | 74 | 68 | 47 | 44 | 39 | 41 | 48 | 43 |
| Neutral | 3 | 3 | 3 | 1 | 1 | 1 | 18 | 18 | 24 |
| Disagree | 7 | 7 | 6 | 35 | 38 | 4 | 1 | 1 | 1 |
| Strongly disagree | 2 | 2 | 2 | 0 | 0 | 0 | 1 | 1 | 1 |

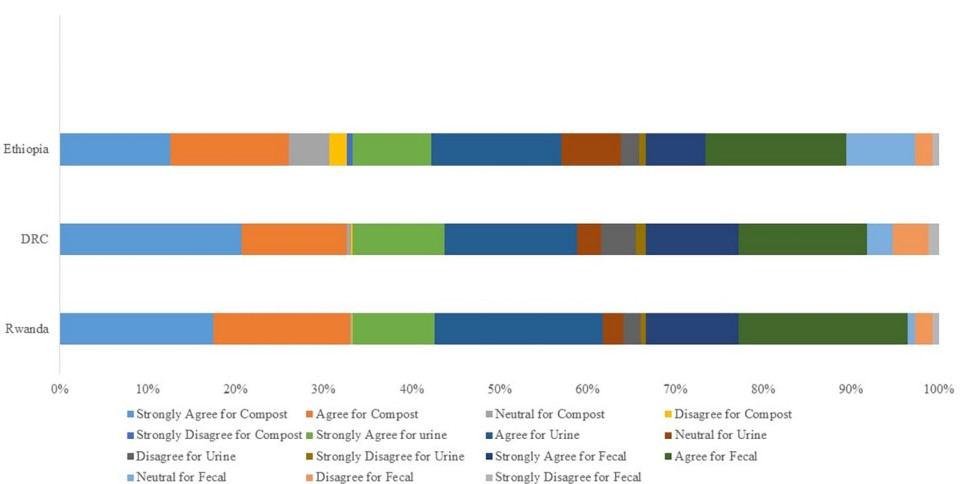

**Fig 2. Household opinion on eating foods grown on CBE fertilizers.**

disagree, and strongly disagree), for each country, considering all respondents (100%) from a given country. We use the percentage scale to enable comparability across countries. The proportion for each opinion along the country scale represents the percentage of respondents in such a country with such an opinion, out of all the total number of respondents (100%). A given respondent could have different opinions across different types of CBE foods.

## 3.2. Regression results

### 3.2.1. Which factors do explain communities' CBE awareness, knowledge, and support?.

Results in Table 5 indicate that households that lived farther from major towns, were less likely by a likelihood of 0.6% to strongly agree to be aware of CBE practices. On the other hand, households whose heads used mobile phones or were married, or had larger family sizes, or had higher incomes, or were older, were more likely by a likelihood of 4.8%, 5.2%, 0.8%, 0.4%, and 0.2% respectively to strongly agree to be aware of CBE practices. Considering country specific effects, households in Rwanda and Ethiopia compared to those in DRC, were more likely by a likelihood of 13% and 46% respectively to strongly agree to be aware of CBE practices.

Results in Table 6, indicate that households that lived farther from towns, or those that suffered severe food insecurity were less likely by a likelihood of 0.2%, and 0.5% respectively, to strongly agree to be knowledgeable about CBE practices. On the other hand, households with more educated heads, or those with larger family sizes, or had higher agricultural income, or older heads, or married heads were more likely by a likelihood of 0.6%, 0.7%, 0.2%, 0.3%, and 5% respectively to strongly agree to be knowledgeable about CBE practices. However, households in Rwanda and Ethiopia compared to those in DRC were more likely by a likelihood of 11%, and 23% respectively to strongly agree to be knowledgeable about CBE practices.

From Table 7, results indicate that households that lived farther from towns, or those that suffered severe food insecurity, were less likely by a likelihood of 1%, and 0.4% respectively to strongly agree to support CBE practices. However, households whose heads used mobile phones, or had higher agricultural incomes, or were older, were more likely by a likelihood of 4.7%, 0.4%, and 0.3% respectively to strongly agree to support CBE practices. Analysis of country effects, indicated that households in Rwanda and Ethiopia, compared to those in DRC, were more likely by a likelihood of 19%, and 42% respectively to strongly agree to support CBE practices.

**Table 5. Marginal effects after generalized ordered logit for awareness on CBE practices.**

| Model Outcomes: Stated Awareness levels | Explanatory Variables | | | | | | | | | Country | |
|---|---|---|---|---|---|---|---|---|---|---|---|
| | Distance to major town (km) | Male household head (D) | Education of head (years) | Mobile phone use (D) | Household size (Persons) | Agric. income /season (USD) | Age of head (years) | Severe food insecurity (score) | Household head is married (D) | Rwanda | Ethiopia |
| (1) **Strongly agree** $Pr = 0.175^{***}$ | **-0.006**[***] | -0.024 | 0.002 | **0.048**[**] | **0.008**[**] | **0.004**[***] | **0.002**[***] | -0.013 | **0.052**[**] | **0.128**[***] | **0.461**[***] |
| | **(0.001)** | (0.023) | (0.002) | **(0.021)** | **(0.004)** | **(0.001)** | **(0.001)** | (0.010) | **(0.024)** | **(0.012)** | **(0.032)** |
| (2) **Agree** $Pr = 0.723^{***}$ | **0.005**[***] | -0.004 | **0.006**[***] | **-0.06**[**] | -0.003 | -0.001 | **-0.002**[**] | 0.005 | 0.003 | **-0.26**[***] | **-0.62**[***] |
| | **(0.002)** | (0.027) | **(0.002)** | **(0.023)** | (0.004) | (0.001) | **(0.001)** | (0.011) | (0.031) | **(0.019)** | **(0.034)** |
| (3) **Neutral** $Pr = -0.22^{***}$ | **0.003**[**] | **0.024**[**] | -0.0002 | 0.015 | -0.001 | **-0.001**[**] | -4x10⁻⁵ | **0.017**[***] | **-0.045**[**] | **1.001**[***] | **1.122**[***] |
| | **(0.001)** | **(0.011)** | (0.001) | (0.026) | (0.002) | **(0.001)** | (0.0004) | **(0.006)** | **(0.019)** | **(0.029)** | **(0.027)** |
| (4) **Disagree** $Pr = 0.301$ | -0.001 | -0.006 | -0.006 | 0.008 | 0.0001 | 0.0001 | -7x10⁻⁶ | 0.125 | -0.006 | -1.026 | -1.414 |
| | (0.009) | (0.723) | (0.0467) | (41.20) | (0.341) | (0.141) | (0.071) | (12.70) | (0.340) | (10.64) | (22.06) |
| (5) **Strongly disagree** $Pr = 0.019$ | -0.0001 | 0.008 | -0.001 | -0.012 | -0.004 | -0.002 | -0.001 | -0.135 | -0.004 | 0.151 | 0.453 |
| | (0.009) | (0.723) | (0.047) | (41.20) | (0.341) | (0.141) | (0.073) | (12.70) | (0.340) | (1.129) | (12.95) |

Standard errors in parentheses

*** p<0.01

** p<0.05

* p<0.1

Pr. is probability for the given outcome when all predictors are at mean values; D is dummy, USD is United States Dollars, KM is kilometers; Approximated likelihood ratio test of proportionality of odds across response categories after ordered logit, Chi2 = 621.52***; Generalized Ordered Logit estimates: Pseudo R2 = 0.396, Chi2 = 1,865.54***, Observations = 2,175; **in bold are significant variables at 5% level and above.**

**Table 6. Marginal effects after generalized ordered logit for knowledge on CBE practices.**

| Model outcomes: Stated Knowledge levels | Explanatory Variables | | | | | | | | | Country | |
|---|---|---|---|---|---|---|---|---|---|---|---|
| | Distance to major town (km | Male household head (D) | Education of head (years) | Mobile phone use (D) | Household size (Person) | Agric. income/ season (USD) | Age of head (years) | Severe food insecurity (score | Household head is married (D) | Rwanda | Ethiopia |
| (1) **Strongly agree** $Pr = 0.152^{***}$ | -0.002* | -0.019 | **0.006**[***] | 0.021 | **0.007**[***] | **0.002**[***] | **0.003**[***] | **-0.005**[**] | **0.053**[***] | **0.109**[***] | **0.233**[***] |
| | (0.001) | (0.015) | **(0.001)** | (0.014) | **(0.002)** | **(0.0003)** | **(0.0004)** | **(0.002)** | **(0.014)** | **(0.014)** | **(0.019)** |
| (2) **Agree** $Pr = 0.625^{***}$ | -0.0003 | -0.003 | **0.001**[***] | 0.006 | **0.001**[**] | **0.0004**[***] | **0.001**[***] | -0.001* | **0.024**[**] | **0.157**[***] | **0.118**[***] |
| | (0.0002) | (0.002) | **(0.0004)** | (0.005) | **(0.001)** | **(0.0001)** | **(0.0002)** | (0.001) | **(0.010)** | **(0.024)** | **(0.026)** |
| (3) **Neutral** $Pr = 0.069^{***}$ | 0.001* | 0.005 | **-0.002**[***] | -0.006 | **-0.002**[**] | **-0.001**[***] | **-0.001**[***] | **0.001**[**] | **-0.017**[***] | **-0.055**[***] | **-0.083**[***] |
| | (0.0003) | (0.004) | **(0.0004)** | (0.004) | **(0.001)** | **(9x10⁻⁵)** | **(0.0001)** | **(0.001)** | **(0.005)** | **(0.008)** | **(0.008)** |
| (4) **Disagree** $Pr = 0.143^{***}$ | 0.001* | 0.015 | **-0.005**[***] | -0.019 | **-0.01**[***] | **-0.002**[***] | **-0.002**[***] | **0.004**[**] | **-0.054**[***] | **-0.193**[***] | **-0.247**[***] |
| | (0.001) | (0.012) | **(0.001)** | (0.013) | **(0.002)** | **(0.0003)** | **(0.0004)** | **(0.002)** | **(0.017)** | **(0.027)** | **(0.023)** |
| (5) **Strongly disagree** $Pr = 0.011^{***}$ | 0.0001 | 0.002 | **-0.001**[***] | -0.002 | **-0.001**[**] | **-0.0002**[***] | **-0.0002**[**] | 0.0004* | **-0.006**[***] | **-0.018**[***] | **-0.021**[***] |
| | (9x10⁻⁵) | (0.001) | **(0.0001)** | (0.001) | **(0.0002** | **(4x10⁻⁵)** | **(6x10⁻⁵)** | (0.0002) | **(0.002)** | **(0.005)** | **(0.005)** |

Standard errors in parentheses

*** p<0.01

** p<0.05

* p<0.1

Pr. is probability for the given outcome when all predictors are at mean values; D is dummy, USD is United States Dollars, KM is kilometers; Approximated likelihood ratio test of proportionality of odds across response categories, after ordered logit, Chi2 = 576.01***; Generalized Ordered Logit estimates: Pseudo R2 = 0.379, Chi2 = 1,787.41***, Observations = 2,175, **in bold are significant variables at 5% level and above.**

In brief, households that stayed farther from towns, or those that suffered severe food insecurity were significantly less likely to strongly agree to be aware, knowledgeable, or supportive of CBE practices. The opposite was true for households whose heads were married, better educated, older, used mobile phones, or had better agricultural income. At country level, Rwandan and Ethiopian households compared to those in DRC, were more likely to be aware, knowledgeable, and supportive of CBE practices–with greater effects being observed in Ethiopia.

**3.2.2. What factors explain households' opinion of "strongly agree" to eat CBE foods?.** From Table 8, households that lived farther from towns were less likely by a likelihood of 0.7% to strongly agree to eat food grown on compost. However, households whose family sizes were bigger, or those whose heads used mobile phones, or were older, were more likely by a likelihood of 0.9%, 6%, and 0.2% respectively to strongly agree to eat foods grown on compost. Country effects indicate that Rwandan and Ethiopian households compared to DRC ones, were less likely by 18%, and 28% respectively to strongly agree to eat foods grown on compost.

From Table 9, households that lived farther from towns, or those which suffered severe food insecurity, or those whose heads were more educated, were less likely by a likelihood of 0.9%, 3.1%, and 0.5% respectively to strongly agree to eat foods grown on treated urine. On the other hand, households with larger family sizes, or those whose heads used mobile phones, or had higher agricultural income, or their heads were older, or married, were more likely by a likelihood of 0.7%, 6%, 0.3%, 0.3%, and 5.5% respectively to strongly agree to eat foods grown on treated urine. However, households in Rwanda and Ethiopia compared to those in DRC, were less likely by a likelihood of 19%, and 15% respectively to strongly agree to eat foods grown on treated urine.

**Table 7. Marginal effects after generalized ordered logit for support on CBE practices.**

| Model Outcomes: Stated Support levels | Explanatory Variables | | | | | | | | | Country | |
|---|---|---|---|---|---|---|---|---|---|---|---|
| | Distance to major town (km | Male household head (D) | Education of head (years) | Mobile phone use (D) | Household size (Person) | Agric. income/ season (USD) | Age of head (years) | Severe food insecurity (score) | Household head is married (D) | Rwanda | Ethiopia |
| (1) **Strongly agree** Pr = 0.178*** | **-0.01**\*\*\* | -0.024 | -0.003 | **0.047**\*\* | 0.006 | **0.004**\*\*\* | **0.003**\*\*\* | **-0.037**\*\*\* | 0.040 | **0.194**\*\*\* | **0.415**\*\*\* |
| | **(0.002)** | (0.024) | (0.002) | **(0.022)** | (0.004) | **(0.001)** | **(0.001)** | **(0.012)** | (0.025) | **(0.019)** | **(0.034)** |
| (2) **Agree** Pr = 0.695*** | **0.005**\*\*\* | -0.020 | **0.009**\*\*\* | **-0.06**\*\* | -9x10^-5 | -0.001 | **-0.001**\*\* | **0.027**\*\* | 0.011 | **-0.33**\*\*\* | **-0.640**\*\*\* |
| | **(0.002)** | (0.025) | **(0.002)** | **(0.024)** | (0.004) | (0.001) | **(0.001)** | **(0.013)** | (0.031) | **(0.023)** | **(0.036)** |
| (3) **Neutral** Pr = -0.19*** | 0.002 | **0.043**\*\*\* | **0.001** | 0.013 | **-0.008**\*\*\* | **-0.002**\*\*\* | -6x10^-5 | **0.019**\*\*\* | -0.039* | **1.024**\*\*\* | **1.193**\*\*\* |
| | (0.001) | **(0.012)** | (0.002) | (0.023) | **(0.003)** | **(0.001)** | (0.001) | **(0.007)** | (0.022) | **(0.021)** | **(0.024)** |
| (4) **Disagree** Pr = 0.303 | -0.002 | -0.005 | -0.006 | 0.003 | 0.004 | 0.0003 | -0.0002 | 0.102 | -0.003 | -1.039 | -1.427 |
| | (0.002) | (0.569) | (0.085) | (32.18) | (0.163) | (0.096) | (0.074) | (11.46) | (0.787) | (13.18) | (28.19) |
| (5) **Strongly disagree** Pr = 0.015 | -2x10^-5 | 0.006 | -0.001 | -0.002 | -0.002 | -0.001 | -0.001 | -0.111 | -0.009 | 0.147 | 0.460 |
| | (0.002) | (0.569) | (0.085) | (32.18) | (0.163) | (0.096) | (0.074) | (11.46) | (0.787) | (0.887) | (16.62) |

Standard errors in parentheses

\*\*\* p<0.01

\*\* p<0.05

\* p<0.1

Pr. is probability for the given outcome when all predictors are at mean values; D is dummy, USD is United States Dollars, KM is kilometers; Approximated likelihood ratio test of proportionality of odds across response categories, after ordered logit, Chi2 = 680.71\*\*\*; Generalized Ordered Logit estimates: Pseudo R2 = 0.338, Chi2 = 1,983\*\*\*, Observations = 2,175, **in bold are significant variables at 5% level and above.**

**Table 8. Marginal effects after generalized ordered logit on eating foods grown on compost (from household, food, and agricultural residues like crop residues and manure).**

| Model Outcomes: Could accept eat Compost grown food | Distance to major town (km | Male household head (D) | Education of head (years) | Mobile phone use (D) | Household size (Person) | Agric. income/ season (USD) | Age of head (years) | Severe food insecurity (score) | Household head is married (D) | Rwanda | Ethiopia |
|---|---|---|---|---|---|---|---|---|---|---|---|
| | | | | | | | | | | Country | |
| (1) **Strongly agree** *Pr = 0.512***** | **-0.007*** | -0.004 | 0.001 | **0.061**** | **0.009**** | 0.001 | **0.002*** | -0.014 | 0.024 | **-0.18*** | **-0.280*** |
| | **(0.001)** | (0.029) | (0.002) | **(0.029)** | **(0.004)** | (0.001) | **(0.001)** | (0.014) | (0.035) | **(0.034)** | **(0.032)** |
| (2) **Agree** *Pr = 0.421***** | **0.005*** | 0.010 | 0.001 | **-0.06**** | -0.005 | 0.001* | **-0.002*** | 0.017 | -0.011 | **0.168*** | **0.079**** |
| | **(0.001)** | (0.030) | (0.002) | **(0.029)** | (0.004) | (0.001) | **(0.001)** | (0.015) | (0.035) | **(0.034)** | **(0.034)** |
| (3) **Neutral** *Pr = 0.041***** | **0.003*** | 0.001 | **-0.003*** | -0.007 | **-0.004**** | **-0.002*** | $-3\times10^{-5}$ | -0.004 | -0.018 | 0.009 | **0.103*** |
| | **(0.001)** | (0.013) | **(0.001)** | (0.011) | **(0.002)** | **(0.001)** | (0.0003) | (0.008) | (0.014) | (0.007) | **(0.020)** |
| (4) **Disagree** *Pr = 0.010* | -0.001 | -0.007 | -0.001 | -0.011 | -0.007 | 0.001 | -0.0004 | -0.011 | 0.024 | -0.003 | -0.019 |
| | (0.001) | (0.011) | (0.001) | (0.012) | (0.005) | (0.001) | (0.0003) | (0.008) | (0.039) | (0.004) | (0.078) |
| (5) **Strongly disagree** *Pr = 0.016**** | -0.0002 | -0.001 | 0.001 | **0.018**** | 0.007 | -0.001 | 0.0001 | **0.012**** | -0.019 | $2\times10^{-7}$ | 0.118 |
| | (0.001) | (0.009) | (0.001) | **(0.008)** | (0.005) | (0.001) | (0.0003) | **(0.005)** | (0.040) | (0.0002) | (0.076) |

Standard errors in parentheses

*** p<0.01

** p<0.05

* p<0.1

Pr. is probability for the given outcome when all predictors are at mean values; D is dummy, USD is United States Dollars, KM is kilometers; Approximated likelihood ratio test of proportionality of odds across response categories, after ordered logit, Chis2 = 183.89***; Generalized Ordered Logit estimates: Pseudo R2 = 0.099, Chi2 = 410.15***, Observations = 2,160, **in bold are significant variables at 5% level and above.**

From Table 10, households that lived farther from towns, or those that suffered severe food insecurity, or those whose heads were more educated, were less likely by a likelihood of 0.9%, 3.4%, and 0.7% respectively to strongly agree to eat foods grown on faecal matter. On the other hand, households whose heads used mobile phones, or were older, or had higher agricultural incomes, or were married, were more likely by a likelihood of 4.8%, 0.3%, 0.3%, and 6.3% respectively, to strongly agree to eat foods grown on faecal matter. Again, households in Rwanda, and Ethiopia were less likely by a likelihood of 17% and 23% respectively to strongly agree to eat foods grown on faecal matter, compared to those in DRC.

In brief, households that lived farther from towns, and those that suffered severe food insecurity, or had more educated heads, were significantly less likely to strongly agree to eat CBE foods especially those grown on treated urine and faeces. On the other hand, households that had larger family sizes, with heads who used mobile phones, or were older, or married, or had higher agricultural incomes, were significantly more likely to eat CBE foods. However, households in Rwanda and Ethiopia compared to those in DRC, were significantly less likely to eat CBE foods. These country effects were larger in magnitude for Ethiopia than Rwanda. In Fig 3, we present graphically the results for the strongly agree option across each of the outcome variables including awareness, knowledge, and support of CBE practices, and acceptance to eat food grown on compost, or treated urine, or faecal matter.

## 4. Analysis and discussion

### 4.1 Descriptive results

Across all food systems, farmers practice CBE concepts, yet, massive accumulation of waste is abundant in towns inhabited mostly by consumers [10,11,38,39,42]. After harvesting, farmers

**Table 9. Marginal effects after generalized ordered logit on eating foods grown on treated urine.**

| Model Outcomes: Could eat treated Urine grown food | Explanatory Variables | | | | | | | | | Country | |
|---|---|---|---|---|---|---|---|---|---|---|---|
| | Distance to major town (km) | Male household head (D) | Education of head (years) | Mobile phone use (D) | Household size (Person) | Agric. income/ season (USD) | Age of head (years) | Severe food insecurity (score) | Household head is married (D) | Rwanda | Ethiopia |
| (1) **Strongly agree** Pr = 0.285*** | **-0.009*** | 0.019 | **-0.005** | **0.060** | 0.007* | **0.003*** | **0.003*** | **-0.031** | 0.055* | **-0.19*** | **-0.150*** |
| | **(0.001)** | (0.026) | **(0.002)** | **(0.025)** | (0.004) | **(0.001)** | **(0.001)** | **(0.013)** | (0.029) | **(0.032)** | **(0.032)** |
| (2) **Agree** Pr = 0.507*** | **0.005*** | -0.018 | **0.010*** | -0.035 | -0.002 | 0.001 | **-0.002** | 0.009 | 0.008 | **0.187*** | 0.047 |
| | **(0.001)** | (0.029) | **(0.002)** | (0.028) | (0.004) | (0.001) | **(0.002)** | (0.014) | (0.034) | **(0.034)** | (0.034) |
| (3) **Neutral** Pr = 0.109*** | **0.002** | 0.022 | **-0.004*** | -0.016 | **-0.008*** | **-0.002*** | -0.0002 | **0.019** | -0.018 | 0.008 | **0.122*** |
| | **(0.001)** | (0.016) | **(0.001)** | (0.018) | **(0.003)** | **(0.0004)** | (0.001) | **(0.008)** | (0.021) | (0.019) | **(0.022)** |
| (4) **Disagree** Pr = 0.077*** | **0.002*** | -0.014 | -0.002 | -0.016 | 0.002 | **-0.001*** | **-0.001** | -0.004 | -0.025 | -0.013 | -0.032* |
| | **(0.001)** | (0.016) | (0.001) | (0.015) | (0.002) | **(0.0004)** | **(0.0004)** | (0.007) | (0.019) | (0.021) | (0.019) |
| (5) **Strongly disagree** Pr = 0.023*** | **0.001** | -0.009 | 0.001 | 0.007 | 0.001 | -0.0004 | $2 \times 10^{-5}$ | 0.005 | -0.020 | 0.008 | 0.012 |
| | **(0.001)** | (0.010) | (0.001) | (0.008) | (0.001) | (0.0003) | (0.0003) | (0.004) | (0.014) | (0.013) | (0.012) |

Standard errors in parentheses

*** $p<0.01$

** $p<0.05$

* $p<0.1$

Pr. is probability for the given outcome when all predictors are at mean values; D is dummy, USD is United States Dollars, KM is kilometers; Approximated likelihood ratio test of proportionality of odds across response categories, after ordered logit, Chi2 = 95.94***; Generalized Ordered Logit estimates: Pseudo R2 = 0.059, Chi2 = 315.05***, Observations = 2,160, **in bold are significant variables at 5% level and above.**

leave crop residues on farm to decompose into compost and replenish soil nutrients. However, there are no proper mechanisms used by farmers to guarantee quality (high- water -retaining capacity, dark colour, high aeration, and dark colour)) of such compost. Therefore, the effectiveness of such practices is not clear. Moreover, farmers have certain crops that they prefer on which to apply this compost. This implies that farmers' interests to practice CBE concepts could be driven by farm/crop typologies, but not necessarily that farmers view CBE practices as such a novel model to optimize resource use. Therefore, knowledge components would be important for CBE innovations to solidify community interests in overall CBE practices. Some farmers in Rwanda also used in addition to field residues, other organic wastes as compost (for instance faecal matter sourced from pit latrines) or animal feeds (for instance cassava peels), implying that innovations that valorise organic waste into compost or animal feeds could stand a higher chance of success among farmers, since farmers could provide a ready market for these CBE products.

High levels of awareness, knowledge, and support for CBE practices across all city-regions suggest a high chance for success of CBE based innovations in these city-regions, and others with similar food systems. This is because these attributes are highly correlated within city-regions. For instance, in Rwanda the Spearman correlation for Awareness (A), Knowledge (K) and Support (S) was A-K = 0.88, A-S = 0.85, K-S = 0.87. For DRC: for, A-K = 0.76, A-S = 0.83, K-S = 0.88. In Ethiopia the results were: A-K = 0.68, A-S = 0.51 and K-S = 0.64. However, the extent (level) of this awareness, knowledge, or support is not yet studied, and thus not known. The high proportions of each attribute in Rwanda could be attributed to a government focused on social and economic developments for nearly three decades [29]. However, high proportions of disagreement in DRC for CBE support highlight the need to further investigate the basis of such attitudes because they hamper optimal scaling of CBE innovations. For instance,

**Table 10. Marginal effects after generalized ordered logit on eating foods grown on faecal matter.**

| Model Outcomes: Could eat fecal matter grown food | Distance to major town (km | Male household head (D) | Education of head (years) | Mobile phone use (D) | Household size (Person) | Agric. income/season (USD) | Age of head (years) | Severe food insecurity (score) | Household head is married (D) | Country Rwanda | Ethiopia |
|---|---|---|---|---|---|---|---|---|---|---|---|
| (1) **Strongly agree** Pr = 0.287*** | **-0.009***** | 0.021 | **-0.007***** | 0.048* | 0.003 | **0.003***** | **0.003***** | **-0.034***** | **0.063**** | **-0.17***** | **-0.226***** |
|  | **(0.002)** | (0.026) | **(0.002)** | (0.025) | (0.004) | **(0.001)** | **(0.001)** | **(0.013)** | **(0.029)** | **(0.033)** | **(0.031)** |
| (2) **Agree** Pr = 0.513*** | **0.004***** | 0.019 | **0.011***** | -0.037 | -7x10⁻⁵ | 0.001 | -0.001 | 0.010 | -0.028 | **0.200***** | **0.085**** |
|  | **(0.001)** | (0.029) | **(0.002)** | (0.027) | (0.004) | (0.001) | (0.001) | (0.014) | (0.033) | **(0.034)** | **(0.034)** |
| (3) **Neutral** Pr = 0.098*** | 0.001 | 0.005 | -0.002* | -0.019 | -0.005* | **-0.001***** | -0.0002 | 0.013* | -0.0004 | **-0.047**** | **0.157***** |
|  | (0.001) | (0.013) | (0.001) | (0.016) | (0.003) | **(0.0004)** | (0.001) | (0.007) | (0.016) | **(0.019)** | **(0.023)** |
| (4) **Disagree** Pr = 0.076*** | **0.003***** | **-0.036**** | -0.002* | 0.002 | -5x10⁻⁵ | **-0.002***** | **-0.001***** | 0.004 | -0.015 | -0.002 | -0.027 |
|  | **(0.001)** | **(0.015)** | (0.001) | (0.014) | (0.002) | **(0.0004)** | **(0.0004)** | (0.006) | (0.018) | (0.022) | (0.019) |
| (5) **Strongly disagree** Pr = 0.026*** | **0.001**** | -0.009 | 0.0003 | 0.006 | 0.002 | -0.001* | -4x10⁻⁵ | 0.007* | -0.020 | 0.014 | 0.011 |
|  | **(0.001)** | (0.009) | (0.001) | (0.008) | (0.002) | (0.0003) | (0.0003) | (0.004) | (0.013) | (0.015) | (0.014) |

Standard errors in parentheses

*** p<0.01

** p<0.05

* p<0.1

Pr. is probability for the given outcome when all predictors are at mean values; D is dummy, USD is United States Dollars, KM is kilometers; Approximated likelihood ratio test of proportionality of odds across response categories, after ordered logit, Chi2 = 204.30***; Generalized Ordered Logit estimates: Pseudo R2 = 0.080, Chi2 = 423.12***, Observations = 2,160, **in bold are significant variables at 5% level and above**.

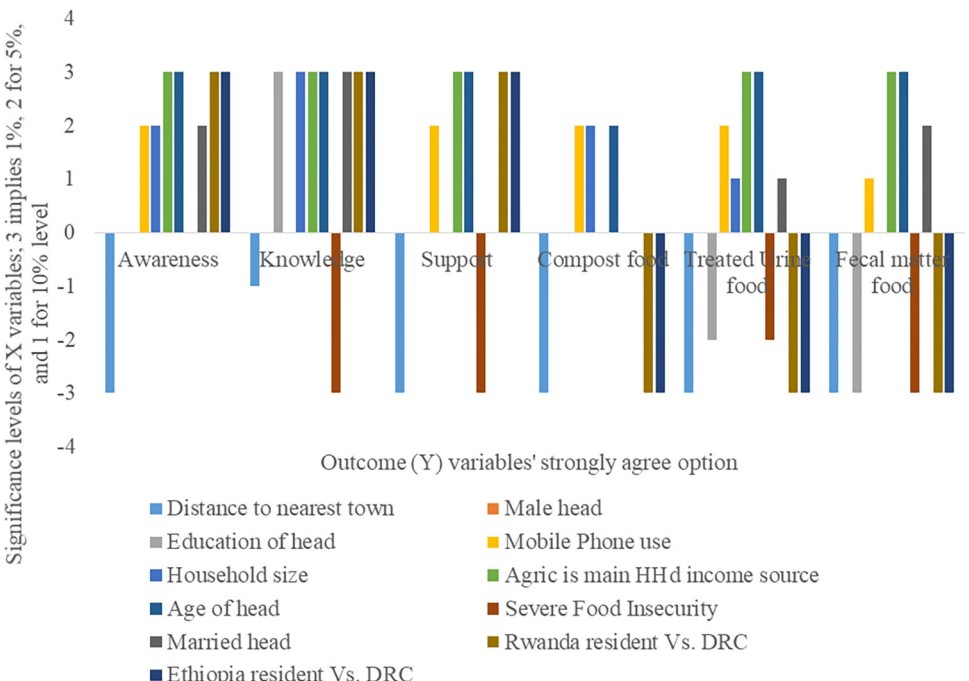

**Fig 3. Significant results for the strongly agree option across all outcome variables.**

there could be an inherent, negative social-cultural perception towards CBE practices among DRC communities, or simply that the instability due to persistent armed civil conflicts in eastern DRC haven't allowed effective community sensitization on CBE practices. Furthermore, understanding reasons behind such a negative opinion could guide appropriate behavioural change interventions. However, notice must be made that our study did not investigate reasons for given consumption opinions. The high possibilities of acceptance to eat CBE foods (Fig 2) could also imply that most consumers would be comfortable eating CBE foods. Moreover, Vogel [43] found that 96% and 86% of farmers in Rwanda would accept organic and human waste recycling practices respectively.

## 4.2 Regression results

The strong relationship between education and CBE knowledge (Table 6), suggests that with proper knowledge, households could understand the usefulness of CBE practices which is essential for scaling out CBE practices. With better education, persons gain better capacity to understand concepts around which livelihoods are sustained [44,45]. This understanding builds trust for such concepts and subsequently acceptance of their products [46]. Alternatively, with better education, households can easily understand the associated health risks around CBE foods from human excreta [47]. Currently, urine and faeces are only stored for some time for instance 3 months before being applied on plants, where farmers assume that this is sufficient time to kill pathogens. Current products have not been tested at all to certify their safety. Our results are similar to those of Wu et al. [48] and McCarthy et al. [49] who found highly educated consumers more willing to eat certified, safer, and healthier foods. Moreover, some foods for instance cassava, it is the roots which directly take-up CBE fertilizers, that are eaten and in case that the CBE fertilizers treatment (processed free from pathogens, heavy metals, pharmaceuticals, and other hazards) was not effective, this could be dangerous to consumers. Hence, to increase acceptance of CBE foods, CBE innovations must be able to prove the safety of CBE products so to motivate consumer confidence, through effective CBE knowledge and learning components. This must also constitute practical user safety guidelines, for safe use of CBE products. Interestingly, for instance the Rwanda Standards Board, has embarked on working with the private sector developing CBE innovations, on the necessary safety standards to be observed before formally availing on the market any CBE products especially those from human excreta which must be used with major safety precautions, unlike the regular compost from green and food waste.

Agriculture is the major livelihood source for most households in studied countries. But because farm inputs are expensive, dependence on agriculture motivates and equips farmers with indigenous knowledge to practice CBE, probably due to a lack of other alternatives. Moreover, within considered countries, recycling of organic waste for reuse in agriculture is largely promoted by governments [28–31,50]. Furthermore, most African food systems rely on naturally available resources and cultivate with minimal farm inputs [51]. Hence, farmers to replenish soil nutrients one natural way used is composting. Thus, CBE innovations could target agricultural households especially those that earn better incomes from agriculture, for demand of CBE products like compost. Households with better agricultural incomes were also more likely to accept eat CBE foods. Our findings are similar to those of Fleischhacker et al. [52] and French et al. [53] who found high-income households to eat healthier and specialized food. CBE foods require resources (time, labour, and money) that may be incurred in recycling, packaging, transportation, and application of CBE fertilizers; thus, CBE foods could currently be comparatively expensive. Therefore, CBE innovations must design and develop cost-effective CBE practices that should yield cost-effective products, and thus CBE foods accessible

to the poor. Nevertheless, we acknowledge that there may be other organic material other than that considered here, which could even be more cost-effective to recycle and reuse–for instance animal (dung) waste, but such other organic waste was not readily available to most households that we interviewed. Also, we acknowledge, that CBE processes currently could be cost-ineffective since compost making requires some costs [54]. However, we believe that if CBE fertilizers are used on high commercial-value crops like vegetables, ornaments, or coffee where high-value products can be generated and sold at competitive global markets, the entire process can ultimately produce net positive revenues. Moreover, composting has been found to have potential for commercial viability [24,55,56]. Since we focussed on exploring existence of CBE practices and community preferences about their products, we do not yet assess CBE commercial viability in the selected city-regions, in this paper, but in the technical report we found that processors sold organic waste for money.

The inverse relationship between increasing household distance to major towns and the likelihood to strongly be aware, knowledgeable, or supportive of CBE practices, or acceptance to eat CBE grown foods, implies that peri-urban and urban consumers would be the more likely market for CBE innovations and CBE foods. CBE practices are intensive in time, labour, and money which could be difficult to apply over relatively larger farm areas as those in rural areas, but handy for averagely small farm areas in peri-urban or urban agriculture. Our findings are similar to those of Mohamad et al. [57] who found urban consumers in Malaysia to be more aware of organic foods, and readily ate such food. Also, the increasingly active sustainability and food safety standards that are more visible in urban settlements, give urban residents more confident in foods from urban markets. Moreover, rural areas are inhabited by communities strong in traditional beliefs, which may hinder acceptance for CBE foods especially that grown on human excreta [58,59]. Also, rural communities are less exposed to daily food insecurity, as these may have more land resources to grow foods [60].

Severe food insecurity was associated with less awareness, knowledge, and support for CBE practices (Tables 5–7). This is in agreement with Nata et al. [61] who found food insecurity to be associated with low support for soil-improving practices, that are similar to CBE practices. Severe food insecurity could partly be leading to the numerous armed conflicts in eastern DRC, with about a hundred waring groups, thus bringing about an environment where knowledge on developmental innovations like CBE can hardly be exchanged [41]. Therefore, CBE innovations could need to prioritize household food security to succeed. For example, CBE innovations could improve animal/crop productivity to directly tackle food insecurity, or yield products that are economically viable to enhance income, and thus improve food security via income pathways. Literature notes that men are more likely to participate in agriculture where there are financial gains [60]. However, composting of from household, food, and agricultural residues like crop residues and manure, the most promoted CBE practice observed, is yet to be reasonably commercialized [28–31,50,62] and could explain the negative likelihood for CBE support among male headed households (Table 7). Thus, CBE innovations should prioritize commercial viability of CBE products to attract male headed (majority) households. Surprisingly, households with severe food insecurity were less likely to eat CBE foods, which could still point to possible perceived health risks given the current methods of producing CBE fertilizers, or merely strong negative social-cultural values, yet these need longer time to change [63]. In some instances, social stigma has been found to negatively influence food acceptance by food insecure persons [64,65]. Therefore, CBE innovations must be aware of potential social dislikes, and strengthen awareness campaigns on safety, and appropriateness of CBE foods. Wrong consumer perceptions may generate social stigma against CBE foods.

Interestingly, we found that married households also had a higher likelihood to be aware and knowledgeable of CBE practices (Tables 5 and 6), and as well accept to eat CBE foods

(Tables 9 and 10). Married households usually have children, hence, are likely to face higher household food insecurity challenges due to having more persons to feed, as did find Kaza et al. [10] and Sulaiman et al. [66]. Hence, resorting to resource conservative foods, such as CBE foods, is logical. Readily available organic waste which makes raw materials cheaper, could enable better access to inputs like compost. For instance, composting has been found to be commercially viable in Asia by Abdul Rahman et al. [55] and Malik and Rawat [56], implying that it is resource use cost-effective. Nevertheless, composting also bares some costs [54]. Unfortunately, we don't ascertain costs related to each CBE practice in this study–given our study's limited scope and data. Our interpretation above is also supported by our results (Tables 6, 8 and 9) where households with larger family sizes were more likely to be knowledgeable of CBE practices, and as well accept to eat CBE foods, especially those grown on compost and treated urine, that have more social acceptability than those grown on faeces. Therefore, CBE innovations could prioritize households with married heads, and those with larger family sizes. Furthermore, households with older heads were also more likely to be aware, knowledgeable, and supportive of CBE practices, and as well accept to eat any kind of CBE foods. As household heads age, they do apparently accumulate knowledge of the benefits of CBE practices, and subsequently accept CBE foods. This is further supported by Babaei et al. [67] who found older household heads (over 45 years) more likely to recycle waste. Thus, CBE innovations could target households with heads of average old working age, as these could even be more likely to accommodate behavioural changes.

More interestingly, our analysis unearths important associations about household heads having mobile phones, and CBE practices and foods. Mobile phones foster awareness, exchange of information, access to knowledge and financial services (remittances and normal payments for especially financially excluded farm households), and maintenance of connectivity among social networks. This is consistent with Sekabira and Qaim [68], Munyegera and Matsumoto [69] and Mukong and Nanziri [70]. Moreover, Kiconco et al. [71] adds that social networks are a formidable source of knowledge, physical and financial support that could be essential in driving acceptance for CBE innovations, and CBE products. Therefore, CBE innovations could prioritize households with mobile phones to enhance information exchange and knowledge delivery.

Finally, the intriguing country effects show that households in Rwanda and Ethiopia were consistently more likely compared to those in DRC to be aware, knowledgeable, and supportive of CBE practices, however the reverse was true regarding accepting to eat CBE foods. This may be explained by the broader political economy aspects. Prior to early 2021 when data for this study was collected, Rwanda and Ethiopia had enjoyed decades of political stability with no civil wars unlike DRC. In fact, eastern DRC, where data was collected, has been suffering from frequent civil wars with over 20,000 United Nations peacekeepers [41]. Such unrests, have disrupted agriculture and other development and bred poverty and hunger, while developmental socio-economic programs were uninterruptedly being rolled-out in Rwanda and Ethiopia [28–31]. However, the higher likelihood of not accepting CBE foods in Rwanda and Ethiopia, could be linked to perceived health risks, or negative culture values.

## 4.3 Study limitations and suggestions for further research

The relatively high consistency of our results in direction and magnitude of effects across city-regions gives us confidence that our results are robust and reliable. Moreover, we used native enumerators who could make logical interpretations of questions in local contexts around such a relatively new CBE concept in surveys. However, using only cross-section data with stated other than observed attributes and opinions is a limitation. We also did not ask respondents reasons for their attributes or opinions. We also exclusively used categorical data for

dependent variables, which may not provide tangible unit effects. Nevertheless, given that CBE literature in Sub Sahara Africa (SSA) is largely scanty, we had to start somewhere to fill the knowledge gap. We recommend that experimental or panel data could be used in future to support this initial research. That said, we are confident that our results reflect the CBE status quo in studied city-regions and can reliably inform successful design, implementation, and scaling of CBE innovations. However, given that this was an exploratory study–with several key aspects around CBE innovations not investigated, based on our findings, we suggest hypotheses to be investigated by future research around CBE innovations as below:

i. Prioritizing knowledge and learning components is essential for uptake of CBE innovations.

ii. Targeting agricultural households guarantees demand for CBE products like compost animal feeds

iii. Prioritizing household food security ensures uptake of CBE innovations

iv. Commercial viability of CBE innovations influences household adoption of CBE practices.

v. Targeting households with heads who use mobile phones, or those of average old working age, or those who are married, or those with better education, enables uptake of CBE innovations.

vi. Ensuring certification, safety, and sustainability standards of CBE products, increases demand for CBE products.

vii. Strengthening awareness on safety standards, and appropriateness of CBE foods for consumption, alleviates potential social and cultural dislikes against CBE foods.

viii. Wrongful consumer perceptions about CBE foods generates social stigma against CBE foods

ix. Enacting comprehensive CBE policies enables faster deployment and scaling of CBE innovations

## 5. Conclusion

Using data from about 2,100 households, we analysed the status quo of CBE in some African food systems. We studied key food commodity value chains (banana in Ethiopia, coffee in DRC, and cassava in Rwanda), to analyse the awareness, knowledge, or support for CBE practices among farmers and consumers in these chains. We assessed opinions of respondents on consumption of CBE foods, and reviewed country policies on CBE practices. Farmers in all countries, and negligible proportions of DRC urban consumers practiced composting. Therefore, nutrient loops are still wide open in the rural-urban nexus. Furthermore, policies concerning circular bioeconomy appear to be insufficient, and thus need immediate attention for formulation, clarity, and enforcement. Nevertheless, our study households were largely aware, knowledgeable, and supportive of CBE practices. Moreover, the majority would accept to eat CBE foods and thus potential demand for CBE foods could exist especially in towns–more so higher income consumers, that would also be interested in organic foods. However, CBE innovations would benefit from the incorporation of a strong learning component to deliver knowledge, primarily targeting households with older or married heads or those with better education, or those that own mobile phones for ease of knowledge transfer. A focus on rural households heavily reliant on agriculture (with better agricultural incomes) as a market for compost, and urban consumers as market for CBE foods could render CBE innovations

economically viable. However, to facilitate cheaper return of compost to rural areas, bulky fresh organic waste processing must happen near dumpsites in towns, and lighter dry processed compost returned to rural areas. CBE innovations could also need to prioritize household food security to attract sustainable community, and government support for CBE practices. CBE policies must be urgently and comprehensively formulated to guide proper design, development, and implementation of CBE innovations.

## Supporting information

**S1 Table. Regulatory initiatives or policies on CBE practices across countries.** (DOCX)

**S1 File. Comprehensive survey protocol used to collect data.** (PDF)

**S2 File. Data used to generate the model results of this paper.** (DTA)

## Acknowledgments

The authors extend their gratitude to all RUNRES scientists, enumerators, and staff of RUNRES partners that have contributed to data collection, and technical guidance in writing this article.

## Author Contributions

**Conceptualization:** Haruna Sekabira.

**Data curation:** Haruna Sekabira, Elke Nijman, Abayneh Feyso, Byamungu Kigangu.

**Formal analysis:** Haruna Sekabira.

**Funding acquisition:** Pius Krütli, Marc Schut, Bernard Vanlauwe, Speciose Kantengwa, Byamungu Kigangu, Johan Six.

**Investigation:** Haruna Sekabira, Elke Nijman.

**Methodology:** Haruna Sekabira.

**Project administration:** Haruna Sekabira, Leonhard Späth, Pius Krütli, Marc Schut, Bernard Vanlauwe, Benjamin Wilde, Kokou Kintche, Speciose Kantengwa, Abayneh Feyso, Byamungu Kigangu, Johan Six.

**Resources:** Haruna Sekabira, Kokou Kintche.

**Software:** Haruna Sekabira.

**Supervision:** Haruna Sekabira, Leonhard Späth, Pius Krütli, Marc Schut, Bernard Vanlauwe, Benjamin Wilde, Kokou Kintche, Speciose Kantengwa, Abayneh Feyso, Johan Six.

**Validation:** Haruna Sekabira, Pius Krütli, Byamungu Kigangu.

**Visualization:** Haruna Sekabira, Benjamin Wilde, Johan Six.

**Writing – original draft:** Haruna Sekabira.

**Writing – review & editing:** Haruna Sekabira, Elke Nijman, Leonhard Späth, Pius Krütli, Marc Schut, Bernard Vanlauwe, Benjamin Wilde, Johan Six.

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
