## [Decision Letter · Decision Letter 0]

9 Aug 2022

PONE-D-22-04453Circular Bioeconomy in African Food Systems: What is the status quo? Insights from Rwanda, DRC, and Ethiopia.PLOS ONE

Dear Dr. Sekabira,

Thank you for submitting your manuscript to PLOS ONE. After careful consideration, we feel that it has merit but does not fully meet PLOS ONE’s publication criteria as it currently stands. Therefore, we invite you to submit a revised version of the manuscript that addresses the points raised during the review process. You'll see that the reviewers have given quite comprehensive comments, especially reviewer 4.   Your response will need to go through all of these carefully before I can consider accepting.

We look forward to receiving your revised manuscript.

Kind regards,

Alison Parker

Academic Editor

PLOS ONE

https://journals.plos.org/plosone/s/file?id=ba62/PLOSOne_formatting_sample_title_authors_affiliations.pdf".

“The authors extend their gratitude to the Swiss Agency for Development and Cooperation (SDC) for funding RUNRES under grant no: 7F09521. The authors are also thankful to all RUNRES scientists, enumerators, and staff of RUNRES partners that have contributed to data collection, and technical guidance in writing this article.”

“JS

grant no: 7F09521

Swiss Agency for Development and Cooperation (SDC)

https://www.eda.admin.ch/eda/en/fdfa/fdfa/organisation-fdfa/directorates-divisions/sdc.html

6. Please upload a copy of Figure 2, to which you refer in your text on page 18. If the figure is no longer to be included as part of the submission please remove all reference to it within the text.

Reviewers' comments:

Reviewer's Responses to Questions

**Comments to the Author**

1. Is the manuscript technically sound, and do the data support the conclusions?

Reviewer #1: Yes

Reviewer #2: Partly

Reviewer #3: Partly

Reviewer #4: Partly

Reviewer #5: Yes

2. Has the statistical analysis been performed appropriately and rigorously? 

Reviewer #1: Yes

Reviewer #2: I Don't Know

Reviewer #3: No

Reviewer #4: I Don't Know

Reviewer #5: Yes

3. Have the authors made all data underlying the findings in their manuscript fully available?

Reviewer #1: Yes

Reviewer #2: Yes

Reviewer #3: No

Reviewer #4: Yes

Reviewer #5: Yes

4. Is the manuscript presented in an intelligible fashion and written in standard English?

Reviewer #1: Yes

Reviewer #2: Yes

Reviewer #3: Yes

Reviewer #4: Yes

Reviewer #5: Yes

5. Review Comments to the Author

Reviewer #1: The manuscript deals with the study on „Circular Bioeconomy in African Farming Systems: What is the status quo? Insights from Rwanda, DRC, and Ethiopia”. The presented topic is of high professional and practical interest bringing a significant added value to potential target group of readers. Based on the research contribution this study can be justified as innovative.

The title reflects the objective and content of the paper.

The abstract clearly indicates background, result, conclusion and implications of key findings. The keywords are adequate.

Introduction of the manuscript is properly designed and combined with a sufficient critical literature review part. What readers require is, by convinced literature review, to understand the clear thinking/consideration why the proposed approach can reach more convinced results. In addition, authors used updated references. The introduction section describes the main purpose, research methodology and the model of the study Key research questions and the the structure of the study are clearly described.

The methodology applied seems to be satisfactory. The binary nature of variables has been avoided and authors have re-analyzed data with proper ordinal scales (strongly agree, agree, neutral, disagree, and strongly disagree).The description of the three variables (aware, knowledgeable, supportive) is clear. The term awareness and knowledge are used in a broader view in the exploratory study. Authors have re-estimated all models with an ordinal regression considering all outcomes’ categories. The authors have used both the ordered logistic regression and the generalized ordered logistic regression to estimate the model. The model fits for the interpreted generalized ordered logit model are also presented in the revised version. Furthermore, authors ran the generalized ordered logit model to correct for the violation of the assumption and the GOL also controls for multi-collinearity. Chi2 values and Pseudo R2 for validation of the model is also presented correctly. Extreme outliers were followed up with the respondents and corrected. For the normal outliers authors have used the box-cox transformations to generate normalized variables. Software STATA/SE 16.0 has been used for the model.

The manuscript reports the results grounded in a statistical sense. Comparing and combining results with previous studies are included as well.

Analysis and discussion part including limitations of the study is well written.

Conclusions reflect all main findings appropriately. In this section the main ideas of the manuscript are presented, the obtained results and their novelty are demonstrated. The value added of this part is presented by the clear analytical methodology used for the analysis by giving the study an original nature.

Reviewer #2: The article is too broad and too long. I am concerned about its relevance. You should have focused more.

The article is too broad and too long. I am concerned about its relevance. You should have focused more.

Reviewer #3: It is an interesting study but the authors lack certain adjustments to make in the text so that the work will be comprehensive to the readers and so that the article will be accepted.

Abstract

I think it is necessary to list in the summary all the statistical instruments used throughout the research and not only talk about logistic regression. It's like it was the only statistical method used.

Introduction

What kind of organic residue are the authors referring to? It's a little confusing in the text. Should these organic residues they are talking about be used only for cassava, banana and coffee crops? This part is not very clear in the introduction and other parts of the text.

I suggest that the authors create after the introduction, a section for the circular economy, the bioeconomy and then for the circular bioeconomy before the work methodology section.

Methodology

In the methodology, the authors have not demonstrated through statistical calculations how they obtained the samples of interviewees in each country. They did not specify whether in each country the sample was probability or not. This may confuse the reader. It must clearly state in the method the categories of people who responded to the questionnaires. Also, tell each country how long it took to collect the data.

Why did you opt for data analysis using descriptive analysis and logistic regression? What is the importance of using this statistical tool?

Are the people who collected the data specialists, students, and professors? It is important to say so in the methodology.

Result

The percentages of the different respondents in each country quoted above in Table 4 do not appear in the latter. The authors need to review this part.

Figure 1 mentioned by the authors does not appear in the text. I suggest a compilation of tables 5 to 10, presenting only the main results.

Analysis and discussions

I suggest that the authors discuss the different categories, CBE practices; awareness, knowledge, and support for CBE practices; consumers’ opinions, explored in their tabular results, adding the different observations for each country.

As an example, authors can consult the article by Mourad (2016).

Mourad, Marie (2016). Recycling, recovering and preventing “food waste”: competing solutions for food systems sustainability in the United States and France. Journal of Cleaner Production, (), S0959652616301536–. doi:10.1016/j.jclepro.2016.03.084

Conclusions

Acceptable.

Reviewer #4: The authors present the results of a suite of surveys and interviews on the uptake and acceptance of circular bioeconomy practices in three African countries, conducted with stakeholders along three different value chains. Having myself conducted interviews on this topic, I am fully aware of the sheer amount of time and effort that is needed to go through the process of ethics approval, getting participant consent, preparing and conducting surveys and interviews, and analysing the results. Doing this across multiple countries that sport a rich variety of languages, and with this number of participants, really is an achievement that is to be commended.

While the results of this endevour are absolutely worthwhile of being published, I feel there are a few challenges to be overcome before the manuscript has matured enough to be ready for publication.

1st Challenge – Framing. The work presented in the manuscript is embedded in a larger project with a much wider scope than the part that is presented in the manuscript. However, although the RUNRES project is mentioned in the manuscript, I feel that the manuscript frames the presented 'CBE status quo work' (which I assume was one of many activities in the RUNRES project) as if it was a standalone project. This creates the following problem.

Within the broader scope of a project like RUNRES, I absolutely buy the idea of focusing on selected value chains in selected locations. After all, this follows best practice in sustainability research in that the project integrates researchers and non-researchers in a place-based manner. Zooming in to the part on CBE practices, however, this creates several issues. Most importantly, framing the paper in terms of the status quo in CBE in African food systems, I would expect an analysis of nutrient flows, hot spots for recovery, and key hinders for recovery and reuse. Put differently: what is the recovery potential of different source material, and what is the demand for different production systems? This goes beyond a certain city or a certain value chain. Also, a comparison across countries is questionable if the value chains are different. All in all, the chosen approach makes total sense if seen in the context of the RUNRES project. Yet there are many flaws and gaps if seen as a standalone project to map the status quo in African food systems.

Basically, I see two ways out here. The first being to extend the study by some sort of material flow analysis. This most likely being beyond the scope of the project, I would suggest sticking with the second way out. Namely, to reframe the paper. Provide a background of the larger RUNRES project. Make clear what contribution this paper makes in this broader context. I think the framing here makes a profound difference. With the suggested alternative framing, I believe a reader would be much more forgiving towards flaws and gaps that become an issue with the current framing – these flaws and gaps are of much lesser importance if the work is presented as part of a whole rather than something that stands on its own.

2nd Challenge – Clarity and definition of terminology. Throughout the manuscript, I take issue with the lack of clarity and definition of important concepts and terminology. For instance, in L91 the term "organic waste" seems to imply any type of organic stream that is considered a waste (i.e., crop residues, animal manure, food waste, human manure, etc.). Likewise, in L99 the term "organic compost from recycled waste" also seems to encompass various organic wastes. However, in the survey it seems that human feces are not considered an organic waste since it is presented as a category of its own. Similarly, it seems that "organic compost" seems not to include e.g. composted feces. It seems to me that organic waste rather refers to crop residues, manure and food waste, but not human waste? Also, CBE practices seem not to be fully defined. I think it would be really helpful for the reader if you listed out the source materials you considered, the practices that you considered, and clearly explain what term is used to refer to what. Right now, I feel there is some ambiguity in the use of the terminology, notably "organic waste" and "organic compost", and this risks leading to confusion.

On a more subtle level, I started wondering how much sense it makes to consider CBE practices, e.g. in Table 4 and 7, as one lump category. I would expect that awareness/knowledge/support would vary significantly across different practices (and source of materials). I would assume that CBE practices involving crop residues and animal manure are reasonably well known/supported, whereas the same does not hold true for human waste as source material. If you lump all source material together, these nuances get completely lost. You do represent it separately later in Figure 1, so I am not sure why sometimes it's presented as a lump category "CBE practices" rather than per individual practice. Also, in section 3, there seem to only be regressions for what are reuses as fertilizer. I did not find regressions for reuse of larvae as feed, although this was also addressed in section 4 of the survey. Is this omission on purpose? Perhaps a short note on the rationale could be good to include.

Either way, the manuscript would strongly benefit from being clear which practices and source materials are considered, and which terms are used to refer to what.

3rd Challenge – Methodological clarity and coherence. I found it difficult to follow what exactly was done in terms of survey and interviews etc and which activities supported which research questions and results. For instance, I don't find the survey clearly described in paragraph 2.2 but I find a survey in the supporting information, which is not cross-linked in the text. At the same time, paragraph 2.2 mentions interviews, but there are no details provided in the supporting information. See also comment on L213 below. I think it would be immensely useful to the reader if it was made clearer which activities were undertaken.

4th Challenge – Reader friendliness. I would invite the authors to spend some thought on how to make the paper more visual. A few ideas to start with. Can the different data collection activities be visually linked to different questions – in order to graphically show which activities contributed to answering which questions? In the regression tables (i.e., Table 5 etc.) – how about sending the numbers to the appendix and make a more visual representation in the main manuscript? where you visually indicate which variables are significant and which are not? I think there must be ways to make patterns more salient than with number in tables – It is very difficult to quickly see patterns this way. Also, a graphic that shows the value chain with different points where organic residuals emerge, and with the terminology you use, could be helpful. Perhaps also include the practices that follow after the residuals. E.g., feces can either become fecal compost for agriculture, or it can go through fly larvae and become animal feed.

Overall, once again – there is an immense amount of work behind the research underpinning this manuscript. Yet, the manuscript itself, like a good compost, needs maturation. Notably regarding framing, clarity, coherence, and graphics.

Specific comments:

Introduction & Paragraph 2.1: The introduction seems to be very long. While the theoretical anchoring of the paper in the CE and CBE concepts is valuable, I feel there is a fair bit of redundancy in the text, especially since the CBE concept is taken up once more in paragraph 2.1. I think there is ample scope to consolidate parts of the introduction and paragraph 2.1.

Paragraph 2.3.2: It seems that this paragraph to some extent contains results, notably L286 to 294 to me seem rather like results.

L 49-51: The first paragraph seems to be on a global level. However, the particular statement here about the fate of waste (i.e., uncontrolled dumpsites and rudimentary sanitation facilities) seems to apply primarily to low-income countries and not so much to high and upper-middle-income countries. In fact, the reference Kaza et al. 2018 that you cite seems does provide a nuanced analysis (see e.g. the blue summary box on page 18. I suggest you somehow clarify that the statement here refers in particular to the context of the countries that are the spatial focus of the paper.

L 55: What do you mean by "particular" food system?

L 65: Is utilization restricted to only food waste and agricultural by-products? Wouldn't the scope of utilizing organic waste streams be broader than than and also include nutrients found e.g. in human excreta, wastewater, food processing wastes, etc.?

L73: "Therefore" indicates a logical link between the preceding sentence and the sentence it introduces. I do not fully see this logical link.

L86: What do you mean by "restorative"? Is it the same as "regenerative" or is it conceptually different? I am not sure all readers would have a clear idea of what makes a food system "restorative".

L128: Here, you mention four countries. In the abstract you only mention three of them. In the survey, you also have four countries. Under paragraph 2.2 it's again only three. I see that South Africa was excluded due to data unavailability (of what sort?). Still, I find it a little bit confusing that you sometimes speak of three and sometimes of four countries.

L180: What do you refer to when you say "and their systems"?

L213: Was there an interview guide/protocol that roughly outlined the questions to be asked? When did you administer the survey that is presented in Appendix 1? To the same sample, prior to the interview? Or is it a different sample?

L219: It would be good to mention here that the survey can be found in the Supporting Information. Also, it would be good to provide some information on how you dealt with the issue of multiple languages. In the Supporting Information, there is a Survey in English. But there are dozens of languages spoken across the countries you considered. How did you deal with these many different languages?

L241: "online" what? searches?

L500: "it is the roots which directly ingest CBE fertilizers". As far as I know "ingest" refers to taking something up into the stomach. So, it would apply to humans and animal but not for plants. Or am I misunderstanding something? Do you mean that it is the roots that are ingested directly by humans (rather than say a banana that is not in contact with the CBE fertilizer in the same way than the roots)?

L558: "composting" of what source material? In general? Or agricultural residues (crop residues and manure) in particular?

L625-641: These points are presented under 4.3 Study limitation. However, they seem to be hypotheses / suggestions for further research rather than limitations of the current study.

Figure 1: Hard to follow. What is the 0-100% of what? Presumably number of respondents? Having 5 columns in the legend would be neat. so it would be a matrix 3 (products) x 5 (levels of agreement). Much easier to read. Also, why do the different sources add up to 100%? Is it not possible that one responded agrees for composts, is neutral for urine, and disagrees for fecal matter? So, should it not be 100% per "product"? If the x-axis is percentage of respondents, that is... Either way, some more explanation would be useful.

Responses to reviewers: the way the responses are organised makes it very hard to see what other reviewers commented and how you addressed it. It would be much easier to read if there were, e.g., 2 columns. One with the reviewer comment and one with the responses. I printed on grayscale, so it was almost impossible to see what is a reviewer comment and what is a response.

Reviewer #5: This an interesting topic on assessing the potential of circular bio economies to promote food systems resilience in developing countries. The paper was well done, elucidating the status quo of circular bio economies in selected countries regarding the effects of age, social media and household type on acceptance of cbe. However, there are few comments to be clarified and the paper can be accepted for publication.

Line 154: Out of place

Line 173: Remove comma

Line 186: May you concisely show how you conducted the participatory exercise

Line 188: Remove fullstop

Line 459 – 460 Its not clear which quality aspect of compost are you talking about. You further explain.

Line 500: Please clarify which treatment? Cassava treatment or the fertiliser treatment. Please specify which safety aspect are you exactly referring to (heavy metals or pharmaceuticals?).

Line 501-503 you may also add the importance of practical guidelines on the actual safe use of these products (especially sanitation related products) and how they can increase yields and increase acceptance. Link to the WHO guidelines, USEPA etc.

Line 503-506: To me the major safety aspects come after handling hazardous waste such as human excreta. Please when explaining differentiate on which type of waste or CBE aspect. Composting using food waste and green waste have no major safety implications. The same applies to line 566, social sigma is associated with human excreta.

Line 575: Composting is not economically viable especially when low nutrient feedstocks are used. Compost is an important source of organic to improve soil properties. May you please further explain why was it viable in Asia?

6. PLOS authors have the option to publish the peer review history of their article (what does this mean?). If published, this will include your full peer review and any attached files.

Reviewer #1: No

Reviewer #2: No

Reviewer #3: No

Reviewer #4: No

Reviewer #5: No

---

## [Author Response · Author response to Decision Letter 0]

23 Sep 2022

PONE-D-22-04453: Circular Bioeconomy in African Food Systems: What is the status quo? Insights from Rwanda, DRC, and Ethiopia.

Editor’s comments:

“The authors extend their gratitude to the Swiss Agency for Development and Cooperation (SDC) for funding RUNRES under grant no: 7F09521. The authors are also thankful to all RUNRES scientists, enumerators, and staff of RUNRES partners that have contributed to data collection, and technical guidance in writing this article.” Funding information is all removed from the main text of the paper.

Please remove any funding-related text from the manuscript and let us know how you would like to update your Funding Statement. Currently, your Funding Statement reads as follows: All funding information is removed from the paper and updated in the cover letter for incorporation in the online platform.

“JS

grant no: 7F09521

Swiss Agency for Development and Cooperation (SDC)

https://www.eda.admin.ch/eda/en/fdfa/fdfa/organisation-fdfa/directorates-divisions/sdc.html

The funders had no role in study design, data collection and analysis, decision to publish, or preparation of the manuscript.” Funding information is corrected in the cover letter – kindly update it online.

3. In your Data Availability statement, you have not specified where the minimal data set underlying the results described in your manuscript can be found. PLOS defines a study's minimal data set as the underlying data used to reach the conclusions drawn in the manuscript and any additional data required to replicate the reported study findings in their entirety. All PLOS journals require that the minimal data set be made fully available. For more information about our data policy, please see http://journals.plos.org/plosone/s/data-availability. This is now addressed with minimal anonymity data provided in S2 File.

Upon re-submitting your revised manuscript, please upload your study’s minimal underlying data set as either Supporting Information files or to a stable, public repository and include the relevant URLs, DOIs, or accession numbers within your revised cover letter. For a list of acceptable repositories, please see http://journals.plos.org/plosone/s/data-availability#loc-recommended-repositories. Any potentially identifying patient information must be fully anonymized. Minimal data is now uploaded in S 2 File.

b) If there are no restrictions, please upload the minimal anonymized data set necessary to replicate your study findings as either Supporting Information files or to a stable, public repository and provide us with the relevant URLs, DOIs, or accession numbers. For a list of acceptable repositories, please see http://journals.plos.org/plosone/s/data-availability#loc-recommended-repositories. Data is availed in supporting Information.

We will update your Data Availability statement on your behalf to reflect the information you provide. We now avail all data used to generate econometric results of the paper in data file, S 2 File.

5. Please include your full ethics statement in the ‘Methods’ section of your manuscript file. In your statement, please include the full name of the IRB or ethics committee who approved or waived your study, as well as whether or not you obtained informed written or verbal consent. If consent was waived for your study, please include this information in your statement as well. This full statement is uploaded using a more recent form of 2021, since the approval would expire every 12 months and need renewal.

6. Please upload a copy of Figure 2, to which you refer in your text on page 18. If the figure is no longer to be included as part of the submission please remove all reference to it within the text. This was indeed an oversight and is corrected as a reference to Fig 1

7. Please include captions for your Supporting Information files at the end of your manuscript, and update any in-text citations to match accordingly. Please see our Supporting Information guidelines for more information: http://journals.plos.org/plosone/s/supporting-information. This has now been correctly done, following the journal procedures

Reviewers' comments:

Reviewer #1: The manuscript deals with the study on „Circular Bioeconomy in African Farming Systems: What is the status quo? Insights from Rwanda, DRC, and Ethiopia”. The presented topic is of high professional and practical interest bringing a significant added value to potential target group of readers. Based on the research contribution this study can be justified as innovative.

The title reflects the objective and content of the paper.

The abstract clearly indicates background, result, conclusion and implications of key findings. The keywords are adequate.

Introduction of the manuscript is properly designed and combined with a sufficient critical literature review part. What readers require is, by convinced literature review, to understand the clear thinking/consideration why the proposed approach can reach more convinced results. In addition, authors used updated references. The introduction section describes the main purpose, research methodology and the model of the study Key research questions and the the structure of the study are clearly described.

The methodology applied seems to be satisfactory. The binary nature of variables has been avoided and authors have re-analyzed data with proper ordinal scales (strongly agree, agree, neutral, disagree, and strongly disagree).The description of the three variables (aware, knowledgeable, supportive) is clear. The term awareness and knowledge are used in a broader view in the exploratory study. Authors have re-estimated all models with an ordinal regression considering all outcomes’ categories. The authors have used both the ordered logistic regression and the generalized ordered logistic regression to estimate the model. The model fits for the interpreted generalized ordered logit model are also presented in the revised version. Furthermore, authors ran the generalized ordered logit model to correct for the violation of the assumption and the GOL also controls for multi-collinearity. Chi2 values and Pseudo R2 for validation of the model is also presented correctly. Extreme outliers were followed up with the respondents and corrected. For the normal outliers authors have used the box-cox transformations to generate normalized variables. Software STATA/SE 16.0 has been used for the model.

The manuscript reports the results grounded in a statistical sense. Comparing and combining results with previous studies are included as well.

Analysis and discussion part including limitations of the study is well written.

Conclusions reflect all main findings appropriately. In this section the main ideas of the manuscript are presented, the obtained results and their novelty are demonstrated. The value added of this part is presented by the clear analytical methodology used for the analysis by giving the study an original nature.

We are grateful to have effectively followed the guidance of reviewer #1, and addressed all their concerns as highlighted in their statement above.

Reviewer #2: The article is too broad and too long. I am concerned about its relevance. You should have focused more.

The article is too broad and too long. I am concerned about its relevance. You should have focused more.

We narrow the focus of the article with more precise wording, thus helping us also to reduce the length of the article from an original 12,687 words to…… or from 29 pages to……..

Reviewer #3: It is an interesting study but the authors lack certain adjustments to make in the text so that the work will be comprehensive to the readers and so that the article will be accepted.

We make all recommended adjustments in the text as advised by reviewer #3 to make the work more concise and comprehensive enough for the readers to understand easily. Below we show point-by-point responses on specific text sections highlighted by the reviewer.

Abstract

I think it is necessary to list in the summary all the statistical instruments used throughout the research and not only talk about logistic regression. It's like it was the only statistical method used. Guidance is much appreciated and now incorporated.

Introduction

What kind of organic residue are the authors referring to? It's a little confusing in the text. Should these organic residues they are talking about be used only for cassava, banana and coffee crops? This part is not very clear in the introduction and other parts of the text.

We now make it clear in Foot note 1, that organic residue or waste is just anything waste which is biological – it can be used for any crops, only that our focus was on the dominant crops in the respective countries.

I suggest that the authors create after the introduction, a section for the circular economy, the bioeconomy and then for the circular bioeconomy before the work methodology section.

Indeed, after the introduction – we incorporated a unified section “the circular bioeconomy concept” but therein describe the circular economy, the bioeconomy, and the circular bioeconomy (lines 156 – 176 on tracked manuscript), to levels we feel are sufficient for the readers in an applied framework. Even in the main body of the introduction (lines 65 – 91) we briefly define each of these three concepts. Otherwise, we think if we expand these to independent sections, we may portray the paper as more of a conceptual theoretic one, than an applied one we intend to portray. Also, we fear that we may make the paper even broader and longer, as reviewer 2 has already pointed out. Hence, we would if acceptable to our kind reviewer, be more comfortable leaving this as one section – instead of three different ones.

Methodology

In the methodology, the authors have not demonstrated through statistical calculations how they obtained the samples of interviewees in each country. Statistically, we elaborated that from the available lists of residents in each city-region, we systematically picked respondents at given intervals for instance 5th, 6th, or 7th for the respective country. This helped us avoid potential bias which could be introduced by random pics, more so that local governments provided all information in a pool. They did not specify whether in each country the sample was probability or not. We clearly state that city regions were selected purposively, and respondents systematically but randomly, which is clearly non-probabilistic – line 197 of tracked manuscript. This may confuse the reader. We now clearly state this in the paper, and thanks to our reviewer for this guidance. It must clearly state in the method the categories of people who responded to the questionnaires. This is now stated clearly on lines 218 – 210 of tracked manuscript. Also, tell each country how long it took to collect the data. Clearly stated as 2 months in each of Rwanda and DRC, and 6 weeks in Ethiopia. Why did you opt for data analysis using descriptive analysis and logistic regression? What is the importance of using this statistical tool? We clearly state in line 253 that descriptive analysis was used to enable us to summarize our data in a way easily assessable by our readers. Then in lines 265 – 272, we specifically state that we use the ordered logistic regression because our data was ordinal, but generally using the logistics regression because our data was reflecting discrete choices of households using these CBE practices. Are the people who collected the data specialists, students, and professors? It is important to say so in the methodology. Clearly stated that we used local graduates from local universities that were specifically trained on our data collection tools in lines 224 – 226. Also, this information was in the original manuscript version – but we make it more coherent of where it should be now for easy identification.

Result

The percentages of the different respondents in each country quoted above in Table 4 do not appear in the latter. The authors need to review this part. Perhaps there was a bit of confusion to the reviewer that all figures in Table 4 are percentages, but we add more clarity on this by using the % symbol besides all columns heading of percentages. However, notice should be made that percentages in the text are a sum of certain categories like agree and strongly agree which we mention in the parentheses within text. Hopefully, this makes it clearer now.

Figure 1 mentioned by the authors does not appear in the text. As per the journal guidelines, I inserted the figure at the end of the article content, after references. I suggest a compilation of tables 5 to 10, presenting only the main results. Because the study is exploratory, we didn’t have a clear single treatment explanatory variable to zero down on across all the analyses – so this makes it difficult to single out main results across all explorations. The only alternative (rear alternative) would be to scale down on components of our Y-variable, but because this is also an ordinal scale variable, the reader may prefer to see how each alternative responds to the explored explanatory variables. Fortunately, we present the tables in a way that reduces their load a maximum of 14 lines, which fits on thirds of a page. Hence, we would hope that this justification suffices with our kind reviewer to maintain the current presentation of the tables.

Analysis and discussions

I suggest that the authors discuss the different categories, CBE practices; awareness, knowledge, and support for CBE practices; consumers’ opinions, explored in their tabular results, adding the different observations for each country. This is excellently done in sections 3.1 and 3.2, where at the beginning we describe these practices and extent of communities’ awareness, knowledge and support of the same (opinions); then in section 3.2 we further explore what determines these opinions while we compare results across countries based on country-specific observations supported by the available data.

As an example, authors can consult the article by Mourad (2016).

Mourad, Marie (2016). Recycling, recovering and preventing “food waste”: competing solutions for food systems sustainability in the United States and France. Journal of Cleaner Production, (), S0959652616301536–. doi:10.1016/j.jclepro.2016.03.084. We appreciate the reviewers’ guidance and the approach of Mourad is well considered to the exchange possible to emphasize clarity of our text.

Conclusions

Acceptable.

Reviewer #4: The authors present the results of a suite of surveys and interviews on the uptake and acceptance of circular bioeconomy practices in three African countries, conducted with stakeholders along three different value chains. Having myself conducted interviews on this topic, I am fully aware of the sheer amount of time and effort that is needed to go through the process of ethics approval, getting participant consent, preparing and conducting surveys and interviews, and analysing the results. Doing this across multiple countries that sport a rich variety of languages, and with this number of participants, really is an achievement that is to be commended. We greatly appreciate the reviewers commendation of our efforts and works, and highly interested in using the reviewers guidance to make the paper better.

While the results of this endevour are absolutely worthwhile of being published, I feel there are a few challenges to be overcome before the manuscript has matured enough to be ready for publication. Grateful for the reviewer’s appreciation and guidance which we take into account as we show in green below.

1st Challenge – Framing. The work presented in the manuscript is embedded in a larger project with a much wider scope than the part that is presented in the manuscript. However, although the RUNRES project is mentioned in the manuscript, I feel that the manuscript frames the presented 'CBE status quo work' (which I assume was one of many activities in the RUNRES project) as if it was a standalone project. This creates the following problem.

Within the broader scope of a project like RUNRES, I absolutely buy the idea of focusing on selected value chains in selected locations. After all, this follows best practice in sustainability research in that the project integrates researchers and non-researchers in a place-based manner. Zooming in to the part on CBE practices, however, this creates several issues. Most importantly, framing the paper in terms of the status quo in CBE in African food systems, I would expect an analysis of nutrient flows, hot spots for recovery, and key hinders for recovery and reuse. Put differently: what is the recovery potential of different source material, and what is the demand for different production systems? This goes beyond a certain city or a certain value chain. Also, a comparison across countries is questionable if the value chains are different. All in all, the chosen approach makes total sense if seen in the context of the RUNRES project. Yet there are many flaws and gaps if seen as a standalone project to map the status quo in African food systems.

Basically, I see two ways out here. The first being to extend the study by some sort of material flow analysis. This most likely being beyond the scope of the project, I would suggest sticking with the second way out. Namely, to reframe the paper. Provide a background of the larger RUNRES project. Make clear what contribution this paper makes in this broader context. I think the framing here makes a profound difference. With the suggested alternative framing, I believe a reader would be much more forgiving towards flaws and gaps that become an issue with the current framing – these flaws and gaps are of much lesser importance if the work is presented as part of a whole rather than something that stands on its own. The advice of the reviewer is much appreciated – and representing that true picture of what we are doing – contributing a certain part of knowledge in a larger RUNRES project framework, which actually also covers material flows and demand and supply analyses of these materials, which we don’t consider in this paper for its bulkiness. We adopt the reviewer’s advice of providing a reasonable background of the larger RUNRES project and then clearly state what our contribution is in such a bigger context project.

2nd Challenge – Clarity and definition of terminology. Throughout the manuscript, I take issue with the lack of clarity and definition of important concepts and terminology. For instance, in L91 the term "organic waste" seems to imply any type of organic stream that is considered a waste (i.e., crop residues, animal manure, food waste, human manure, etc.). This is clarified in several foot notes and parentheses where necessary that organic waste is all waste including human waste, however, due to communities lay understanding of organic waste being such without human waste since this is less acceptable or usable to them, we treated human waste independently – which is appropriate informative. Likewise, in L99 the term "organic compost from recycled waste" also seems to encompass various organic wastes. However, in the survey it seems that human feces are not considered an organic waste since it is presented as a category of its own. Similarly, it seems that "organic compost" seems not to include e.g. composted feces. It seems to me that organic waste rather refers to crop residues, manure and food waste, but not human waste? Also, CBE practices seem not to be fully defined. I think it would be really helpful for the reader if you listed out the source materials you considered, the practices that you considered, and clearly explain what term is used to refer to what. This is now excellently done in several parentheses and food notes. Also, when we come to CBE-derived fertilizers in the questions section, we clearly mention that compost referred to is only that without human excreta, but human excreta is considered different because of the perceived low use willingness and general consideration that it is also in general a compost ingredient. Right now, I feel there is some ambiguity in the use of the terminology, notably "organic waste" and "organic compost", and this risks leading to confusion. The listing is now clearly done of what is considered as source materials, and practices considered. In fact, as the reviewer notes in the first statement, to us – organic waste was anything sourced from organic residues including human feces, but because there was gross lack of literature on the subject and communities generally in their usual lives were not considering human feces as useful resource in most cases, this is why we made a secondary follow-up question isolating human waste. Nevertheless, we make this process of thinking clear in the paper now, such that clarity is guanranteed.

On a more subtle level, I started wondering how much sense it makes to consider CBE practices, e.g. in Table 4 and 7, as one lump category. I would expect that awareness/knowledge/support would vary significantly across different practices (and source of materials). I would assume that CBE practices involving crop residues and animal manure are reasonably well known/supported, whereas the same does not hold true for human waste as source material. Exactly, and this is why we had the same assumption – and put it to test, after which we asked about human waste differently because we knew communities treated human waste differently from other organic waste. If you lump all source material together, these nuances get completely lost. You do represent it separately later in Figure 1, so I am not sure why sometimes it's presented as a lump category "CBE practices" rather than per individual practice. In the beginning, yes, it is presented as a lumpsum for the general background but analytically as even in the survey questions, each of these practices was treated differently, thus also informing our discussions and conclusions that way. We could not have avoided starting with the general picture. Also, in section 3, there seem to only be regressions for what are reuses as fertilizer. I did not find regressions for reuse of larvae as feed, although this was also addressed in section 4 of the survey. Is this omission on purpose? Perhaps a short note on the rationale could be good to include. Indeed, in the regressions we do not show the reuse of larvae as feed, yet in the survey we had addressed this – this was because due to the exploratory nature of the study, we included the practice in the survey, but on the ground there were no individual households using the practice at a reasonable scale (intended use regularly), except for some organizations of farmers where farmers worked in groups but not individuals at household level. So we could not run household based regressions on the same. Our hope is that after learning of the possibility and benefits of reuse of larvae as animal feed, that it would be adopted at household level. Again, we mention this now in the paper to enhance clarity.

Either way, the manuscript would strongly benefit from being clear which practices and source materials are considered, and which terms are used to refer to what. We agree with the reviewer, and we address this now.

3rd Challenge – Methodological clarity and coherence. I found it difficult to follow what exactly was done in terms of survey and interviews etc and which activities supported which research questions and results. For instance, I don't find the survey clearly described in paragraph 2.2 but I find a survey in the supporting information, which is not cross-linked in the text. At the same time, paragraph 2.2 mentions interviews, but there are no details provided in the supporting information. See also comment on L213 below. I think it would be immensely useful to the reader if it was made clearer which activities were undertaken. We re-write the methods section with more clarity and clear mentioning of which item supports the other coherently. We also now correctly enlist the survey in the supporting information as S File. We are grateful to the reviewer for bringing this up.

4th Challenge – Reader friendliness. I would invite the authors to spend some thought on how to make the paper more visual. A few ideas to start with. Can the different data collection activities be visually linked to different questions – in order to graphically show which activities contributed to answering which questions? In the regression tables (i.e., Table 5 etc.) – how about sending the numbers to the appendix and make a more visual representation in the main manuscript? where you visually indicate which variables are significant and which are not? I think there must be ways to make patterns more salient than with number in tables – It is very difficult to quickly see patterns this way. Also, a graphic that shows the value chain with different points where organic residuals emerge, and with the terminology you use, could be helpful. Perhaps also include the practices that follow after the residuals. E.g., feces can either become fecal compost for agriculture, or it can go through fly larvae and become animal feed. Indeed, we fully agree with our honorable reviewer that visual indicators can make it clearer for the readers – however, we fear for the space if we would incorporate an indicator for each table, after all we would have to have the tables too at least in the supporting information – yet tables would be more essential to make a good meaning of the data and the presented results. HOWEVER, to strike a balance on these two goals, within each table we bold the significant figures corresponding to various variables and respective opinions, and we also mention this highlight below each table. THEN, in the main background of the papers, as earlier also advised by our honorable reviewer, we include a comprehensive graphic showing ours and the project’s conceptualization of resource use – seriously hoping that these additions serve the two goals of achieving easier visual clarity while we keep the paper less bulky. Nevertheless, we make a graphic summary of the key results for strongly agree on each aspect and present this just before the summary presentation of the results before conclusions.

Overall, once again – there is an immense amount of work behind the research underpinning this manuscript. Yet, the manuscript itself, like a good compost, needs maturation. Notably regarding framing, clarity, coherence, and graphics. This we closely observe across all of the manuscripts -especially regarding framing for clarity, coherence, and use of graphics and visual aids possible

Specific comments:

Introduction & Paragraph 2.1: The introduction seems to be very long. While the theoretical anchoring of the paper in the CE and CBE concepts is valuable, I feel there is a fair bit of redundancy in the text, especially since the CBE concept is taken up once more in paragraph 2.1. I think there is ample scope to consolidate parts of the introduction and paragraph 2.1.

Paragraph 2.3.2: It seems that this paragraph to some extent contains results, notably L286 to 294 to me seem rather like results. We remove section L286 – 294 into the results section because it is indeed from the analyzed data – and as well try to harmonize the wording around CBE in the main part of the introduction and in some instances remove it completely to avoid potential redundancy .

L 49-51: The first paragraph seems to be on a global level. However, the particular statement here about the fate of waste (i.e., uncontrolled dumpsites and rudimentary sanitation facilities) seems to apply primarily to low-income countries and not so much to high and upper-middle-income countries. In fact, the reference Kaza et al. 2018 that you cite seems does provide a nuanced analysis (see e.g. the blue summary box on page 18. I suggest you somehow clarify that the statement here refers in particular to the context of the countries that are the spatial focus of the paper. Thanks for this guidance, and we correctly amend this to do the correct reflection of the least developed countries.

L 55: What do you mean by "particular" food system? We correct this to imply the respective food system where production should have happened

L 65: Is utilization restricted to only food waste and agricultural by-products? Wouldn't the scope of utilizing organic waste streams be broader than than and also include nutrients found e.g. in human excreta, wastewater, food processing wastes, etc.? The latter is correct and is the intended reflection of our paper – we re-emphasize this in the text.

L73: "Therefore" indicates a logical link between the preceding sentence and the sentence it introduces. I do not fully see this logical link. This is corrected and the presumptive link now removed.

L86: What do you mean by "restorative"? Is it the same as "regenerative" or is it conceptually different? I am not sure all readers would have a clear idea of what makes a food system "restorative". Conceptually, we use it to imply that the system can recover for any resource usage, but for more broader clarity – we adopt the reviewers advise for “regenerative” as this brings out the “re-creating” intention better. We actually remove all the added wording to make the statement more concise 

L128: Here, you mention four countries. In the abstract you only mention three of them. In the survey, you also have four countries. Under paragraph 2.2 it's again only three. I see that South Africa was excluded due to data unavailability (of what sort?). Still, I find it a little bit confusing that you sometimes speak of three and sometimes of four countries. This was an oversight, and has been corrected. One country started their processes late, and thus was not considered for the study, we correct it to 3, but mention later that RUNRES is running in 4 countries, although we use data from only 3 countries, which must have ben the reflection in the abstract.

L180: What do you refer to when you say "and their systems"? we remove this wording as could be confusing since indeed we referred to the food systems 

L213: Was there an interview guide/protocol that roughly outlined the questions to be asked? When did you administer the survey that is presented in Appendix 1? To the same sample, prior to the interview? Or is it a different sample?

L219: It would be good to mention here that the survey can be found in the Supporting Information. Also, it would be good to provide some information on how you dealt with the issue of multiple languages. In the Supporting Information, there is a Survey in English. But there are dozens of languages spoken across the countries you considered. How did you deal with these many different languages?

L241: "online" what? searches?

L500: "it is the roots which directly ingest CBE fertilizers". As far as I know "ingest" refers to taking something up into the stomach. So, it would apply to humans and animal but not for plants. Or am I misunderstanding something? Do you mean that it is the roots that are ingested directly by humans (rather than say a banana that is not in contact with the CBE fertilizer in the same way than the roots)?

L558: "composting" of what source material? In general? Or agricultural residues (crop residues and manure) in particular?

L625-641: These points are presented under 4.3 Study limitation. However, they seem to be hypotheses / suggestions for further research rather than limitations of the current study.

Figure 1: Hard to follow. What is the 0-100% of what? Presumably number of respondents? Having 5 columns in the legend would be neat. so it would be a matrix 3 (products) x 5 (levels of agreement). Much easier to read. Also, why do the different sources add up to 100%? Is it not possible that one responded agrees for composts, is neutral for urine, and disagrees for fecal matter? So, should it not be 100% per "product"? If the x-axis is percentage of respondents, that is... Either way, some more explanation would be useful.

Responses to reviewers: the way the responses are organised makes it very hard to see what other reviewers commented and how you addressed it. It would be much easier to read if there were, e.g., 2 columns. One with the reviewer comment and one with the responses. I printed on grayscale, so it was almost impossible to see what is a reviewer comment and what is a response.

Reviewer #5: This an interesting topic on assessing the potential of circular bio economies to promote food systems resilience in developing countries. The paper was well done, elucidating the status quo of circular bio economies in selected countries regarding the effects of age, social media and household type on acceptance of cbe. However, there are few comments to be clarified and the paper can be accepted for publication.

Line 154: Out of place

Line 173: Remove comma

Line 186: May you concisely show how you conducted the participatory exercise

Line 188: Remove fullstop

Line 459 – 460 Its not clear which quality aspect of compost are you talking about. You further explain. RESPONSE: This is now explained with some attributes of quality compost

Line 500: Please clarify which treatment? Cassava treatment or the fertiliser treatment. Please specify which safety aspect are you exactly referring to (heavy metals or pharmaceuticals?).

Line 501-503 you may also add the importance of practical guidelines on the actual safe use of these products (especially sanitation related products) and how they can increase yields and increase acceptance. Link to the WHO guidelines, USEPA etc.

Line 503-506: To me the major safety aspects come after handling hazardous waste such as human excreta. Please when explaining differentiate on which type of waste or CBE aspect. Composting using food waste and green waste have no major safety implications. The same applies to line 566, social sigma is associated with human excreta.

Line 575: Composting is not economically viable especially when low nutrient feedstocks are used. Compost is an important source of organic to improve soil properties. May you please further explain why was it viable in Asia?

---

## [Decision Letter · Decision Letter 1]

5 Oct 2022

Circular Bioeconomy in African Food Systems: What is the status quo? Insights from Rwanda, DRC, and Ethiopia.

PONE-D-22-04453R1

Dear Dr. Sekabira,

We’re pleased to inform you that your manuscript has been judged scientifically suitable for publication and will be formally accepted for publication once it meets all outstanding technical requirements.

Kind regards,

Alison Parker

Academic Editor

PLOS ONE

Additional Editor Comments (optional):

Reviewers' comments:

Reviewer's Responses to Questions

**Comments to the Author**

1. If the authors have adequately addressed your comments raised in a previous round of review and you feel that this manuscript is now acceptable for publication, you may indicate that here to bypass the “Comments to the Author” section, enter your conflict of interest statement in the “Confidential to Editor” section, and submit your "Accept" recommendation.

Reviewer #1: All comments have been addressed

Reviewer #2: (No Response)

Reviewer #5: All comments have been addressed

2. Is the manuscript technically sound, and do the data support the conclusions?

Reviewer #1: Yes

Reviewer #2: (No Response)

Reviewer #5: Yes

3. Has the statistical analysis been performed appropriately and rigorously? 

Reviewer #1: Yes

Reviewer #2: (No Response)

Reviewer #5: Yes

4. Have the authors made all data underlying the findings in their manuscript fully available?

Reviewer #1: Yes

Reviewer #2: (No Response)

Reviewer #5: Yes

5. Is the manuscript presented in an intelligible fashion and written in standard English?

Reviewer #1: Yes

Reviewer #2: (No Response)

Reviewer #5: Yes

6. Review Comments to the Author

Reviewer #1: Along with the revised manuscript, authors have submitted a detailed revision letter as well. Every comment of the reviewer has been addressed. In the revised manuscript authors have clearly indicated all edits. Congratulations!

Reviewer #2: (No Response)

Reviewer #5: The paper was excellently done and meets the minimum standards of the PLOS ONE journal. All the comments have been addressed and this is ready for publication.

7. PLOS authors have the option to publish the peer review history of their article (what does this mean?). If published, this will include your full peer review and any attached files.

Reviewer #1: No

Reviewer #2: No

Reviewer #5: **Yes: **William Musazura

---

## [Editor Report · Acceptance letter]

12 Oct 2022

PONE-D-22-04453R1 

Circular Bioeconomy in African Food Systems: What is the status quo? Insights from Rwanda, DRC, and Ethiopia. 

Dear Dr. Sekabira:

I'm pleased to inform you that your manuscript has been deemed suitable for publication in PLOS ONE. Congratulations! Your manuscript is now with our production department. 

Kind regards, 

on behalf of

Dr. Alison Parker 

Academic Editor

PLOS ONE